# Spark Transformer: Reactivating Sparsity in Transformer FFN and Attention

**Chong You**[*]    **Kan Wu**[*†]    **Zhipeng Jia**[*‡]    **Lin Chen**[*]    **Srinadh Bhojanapalli**
**Jiaxian Guo**    **Utku Evci**    **Jan Wassenberg**    **Praneeth Netrapalli**
**Jeremiah J. Willcock**    **Suvinay Subramanian**    **Felix Chern**    **Alek Andreev**
**Shreya Pathak**    **Felix Yu**    **Prateek Jain**    **Henry M. Levy**    **David E. Culler**
**Sanjiv Kumar**

Google

## Abstract

The discovery of the *lazy neuron phenomenon* [54], where fewer than $10\%$ of the feedforward networks (FFN) parameters in trained Transformers are activated per token, has spurred significant interests in *activation sparsity* for enhancing large model efficiency. While notable progress has been made in translating such sparsity to wall-time benefits across CPUs, GPUs, and TPUs, modern Transformers have moved away from the ReLU activation function crucial to this phenomenon. Existing efforts on re-introducing activation sparsity, e.g., by reverting to ReLU, applying top-$k$ masking or a sparse predictor, often degrade model quality, increase parameter count, complicate training. *Sparse attention*, the application of sparse activation to the attention mechanism, often face similar challenges.

This paper introduces the *Spark Transformer*, a novel architecture that achieves high activation sparsity in both FFN and the attention mechanism while maintaining model quality, parameter count, and standard training procedures. Our method realizes sparsity via top-$k$ masking for explicit control over sparsity level. Crucially, we introduce *statistical top-$k$*, a hardware-accelerator-friendly, linear-time approximate algorithm that avoids costly sorting and mitigates significant training slowdown from standard top-$k$ operators. Furthermore, Spark Transformer reallocates existing FFN parameters and attention key embeddings to form a *low-cost predictor* for identifying activated entries. This design not only mitigates quality loss from enforced sparsity, but also enhances wall-time benefit. Pretrained with the Gemma-2 recipe, Spark Transformer demonstrates competitive performance on standard benchmarks while exhibiting significant sparsity: only $8\%$ of FFN neurons are activated, and each token attends to a maximum of 256 tokens. This translates to a $2.5\times$ reduction in FLOPs, leading to decoding wall-time speedups of up to $1.79\times$ on CPU and $1.40\times$ on GPU.

## 1 Introduction

The machine learning field has experienced a rapid expansion in the scale of Transformer models [4, 3, 27, 1], significantly advancing the state-of-the-art in language understanding and generation. However, this pursuit of larger models is often constrained not by inherent limitations in model quality [44], but by the soaring computational costs [76, 68] associated with the number of parameters. This

---

[*]Equal contribution
[†]Now at xAI
[‡]Now at Anthropic

39th Conference on Neural Information Processing Systems (NeurIPS 2025).

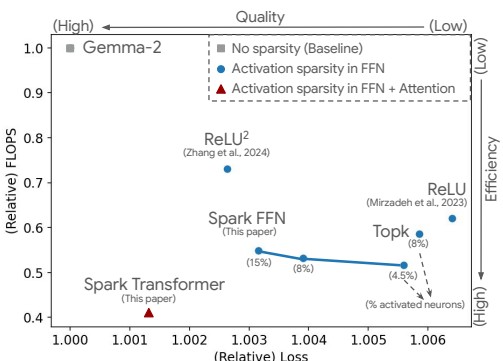

(a) FLOPs per token vs. quality (1/6 of full training)

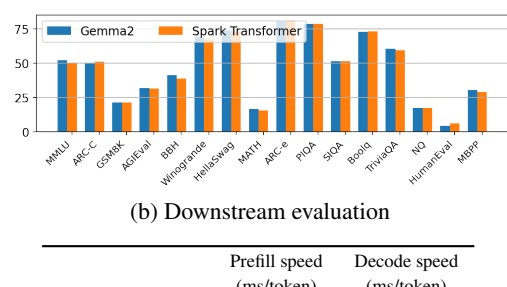

(b) Downstream evaluation

|  | Prefill speed (ms/token) | Decode speed (ms/token) |
|---|---|---|
| Gemma-2 | 28 | 141 |
| Spark Transformer | 15 | 86 |

(c) CPU inference wall time

Figure 1: Spark Transformer improves inference efficiency via activation sparsity in both FFN and attention, while being nearly quality neutral. **(a) Comparison to prior work** in terms of relative FLOPs per token at 8k sequence length (y-axis) vs relative training loss (x-axis)[4]. [■] We use standard Gemma-2 [30] as baseline, which has no activation sparsity. [●] Methods employing activation sparsity within the FFN layers only. *Our Spark FFN achieves the most favorable trade-off compared to ReLU, ReLU$^2$, and Topk*, which refer to standard Gemma-2 with activation function switched to ReLU [64], ReLU$^2$ [92], and the composition of Topk and GELU, respectively. [▲] Combining Spark FFN (with 8% activated parameters) with Spark *Attention* (with at most 256 attended tokens), *our Spark Transformer achieves performance comparable to Gemma-2 while reducing FLOPs to 40%.* **(b) Evaluation on standard downstream tasks** confirms near-quality neutrality of Spark Transformer. **(c) Prefill / decode wall time** demonstrate a $1.86\times/1.64\times$ speedup resulting from FLOPs reduction. Results are obtained on a 4-Core CPUs for prompts of 4096 tokens. For prefill, the prompt is chunked into batches of 64 tokens, following a default setup of gemma.cpp [33].

challenge is compounded by the trend towards models capable of processing increasingly longer input sequences [72], where computational demands scale proportionally with input length.

*Activation sparsity* has emerged as a prominent technique for mitigating the computational burdens associated with both large model sizes and long input sequences. For large models, activation sparsity reduces computational cost by activating only a small fraction of the model's parameters for each input. This approach has attracted considerable attention due to the observed *lazy-neuron* phenomenon [54] where *natural* activation sparsity occurs in the feed-forward networks (FFNs) of traditional Transformer models like T5 [71] and ViT [26], without explicit enforcement [91]. Subsequent work has successfully demonstrated wall-time benefits from activation sparsity on multiple hardware platforms, including CPU [90], GPU [82, 58], and TPU [89].

Despite the great success, a fundamental challenge arises in the application of sparse activation to the latest and state-of-the-art models. That is, the inherent sparsity, which is pivotal for obtaining efficiency, is largely absent due to the adoption of *gated non-ReLU* activation functions [19], widely adopted in, e.g., Mistral [41], Gemma [30], LLAMA [27]. This motivates the following question:

> *Can we re-introduce a high level of activation sparsity in recent Transformers,*
> *without compromising their quality?*

To answer this question, prior work has explored reverting to ReLU variants [64, 92] or incorporating top-$k$ thresholding [89, 81, 83]. However, *(**Challenge #1**) these methods often lead to a degradation in model quality (see Figure 1a).* Furthermore, *(**Challenge #2**) top-$k$ thresholding based methods not only introduce non-differentiability that compromises model quality, but also involve sorting to obtain maximally activated neurons, which is inefficient on ML accelerators, such as TPUs [43], with possibly a $10\times$ training slowdown (see Figure 6).* Finally, a low-cost predictor is often introduced to identify activated parameters, critical for maximizing efficiency benefits [58, 90, 82, 89]. However, *(**Challenge #3**) the inclusion of such a predictor increases training pipeline complexity, incurs additional training costs and parameters, and exacerbates model quality loss.*

---

[4]All models in Figure 1a were trained with standard Gemma-2 recipe for 1/6 of the full pretraining iterations. All other results in this paper are obtained for a fully pretrained Spark Transformer.

Sparse attention, the application of activation sparsity to the attention mechanism, presents similar challenges. A common strategy is top-$k$ attention [35], which applies a top-$k$ mask to the attention coefficients. This can be augmented with a low-cost predictor to further enhance efficiency [74, 88, 50]. Nevertheless, achieving both high sparsity and accurate prediction without resorting to complex procedures, training slowdown from top-$k$, and compromising quality remains an open problem.

**Contributions.** This paper introduces the *Spark Transformer*, a novel architecture that achieves a strong level of activation sparsity in both FFN and attention mechanism with minimal impact on quality, providing a positive answer to the question posed above.

Spark Transformer comprises Spark FFN and Spark Attention, both of which exploit the interpretation of FFNs and attention mechanisms as key-value lookup tables [32] to provide a unified framework for sparsity and a low-cost predictor (see Figure 2). Our **low-cost predictor** is constructed by repurposing a subset of the dimensions of the query and key vectors to compute an importance score for each key-value pair (see Section 2). This design addresses *Challenge #3* by avoiding the introduction of extra parameters, enabling all model parameters to be trained in a single stage.

Our key technical tool for introducing sparsity is a **statistical top-$k$** operator, which is applied to the predicted scores in Spark FFN and Spark Attention to select the activated keys. In particular, statistical top-$k$ is a linear-complexity algorithm for approximate nearest neighbor search, which addresses the issue of high computation cost of standard top-$k$ algorithms (i.e., *Challenge #2*). As explained in Section 3, this is achieved by fitting a Gaussian distribution to the activation scores and estimating a threshold that selects the top entries. Moreover, it enjoys continuously differentiability almost everywhere by the usage of a soft-thresholding operator, making the network more amenable to optimization.[5]

Finally, we demonstrate that Spark Transformer alleviates quality loss (i.e., *Challenge #1*) by performing a full pretraining using the Gemma-2 recipe [30]. Comparison with prior work shows that Spark Transformer exhibits a more favorable trade-off between FLOPs reduction and quality measured by pretraining loss (Figure 1a). Further evaluation on standard benchmarks confirms that Spark Transformer closely matches the performance of Gemma-2 (see Figure 1b), despite exhibiting a high degree of sparsity: only 8% of FFN neurons are activated, and each token attends to at most 256 tokens. Leveraging this sparsity, we assess the model's inference efficiency on CPUs, demonstrating wall-time speedups of $1.86\times$ for prefill and $1.64\times$ for decode (see Figure 1c). Furthermore, an up to $1.4\times$ wall-time speedup is obtained on NVIDIA T4 GPU (see Section 4). This enhanced efficiency broadens access to high-quality models for users with limited access to high-FLOP hardware, such as high-end GPUs and TPUs.

## 2 Spark Transformer

This section describes Spark FFN and Spark Attention, the two components of Spark Transformer.

### 2.1 Spark FFN

FFNs in a standard Transformer are two-layer multi-layer perceptrons that map an input token $q \in \mathbb{R}^{d_{\text{model}}}$ to an output

$$\text{Standard-FFN}(q; K, V) \stackrel{\text{def}}{=} V \cdot \sigma(K^\top q) \in \mathbb{R}^{d_{\text{model}}}. \tag{1}$$

In above, $\{K, V\} \subseteq \mathbb{R}^{d_{\text{model}} \times d_{\text{ff}}}$ are trainable model parameters, and $\sigma()$ is a nonlinear activation function. We ignore the dependency on layer index to simplify the notations.

Each matrix multiplication in eq. (1) has $2d_{\text{model}} \cdot d_{\text{ff}}$ FLOPs hence overall the computation cost is $4d_{\text{model}} \cdot d_{\text{ff}}$. Previous work shows that when $\sigma()$ is ReLU, the activation map $\sigma(K^\top q)$ is very sparse after model training. The sparsity can be used trivially to reduce the computation costs in the calculation of its product with the second layer weight matrix $V$ [54], reducing the overall FLOPs count of FFN to $2d_{\text{model}} \cdot (d_{\text{ff}} + k)$, where $k \ll d_{\text{ff}}$ is the number of nonzero entries in the activation. Note that the sparsity cannot be used to reduce the computation costs associated with $K$, which constitute half of the total FLOPs in FFN.

---

[5]Statistical top-k may also be applicable to Mixture-of-Experts, which face the same non-differentiability and inefficiency challenges in their routing mechanism. A recent paper [87] also used the soft-thresholding in MoE but still relies on a costly sorting operator. Hence, the linear complexity of statistical top-$k$ could address a key computational bottleneck, especially as MoEs trend toward a larger number of experts [18, 36].

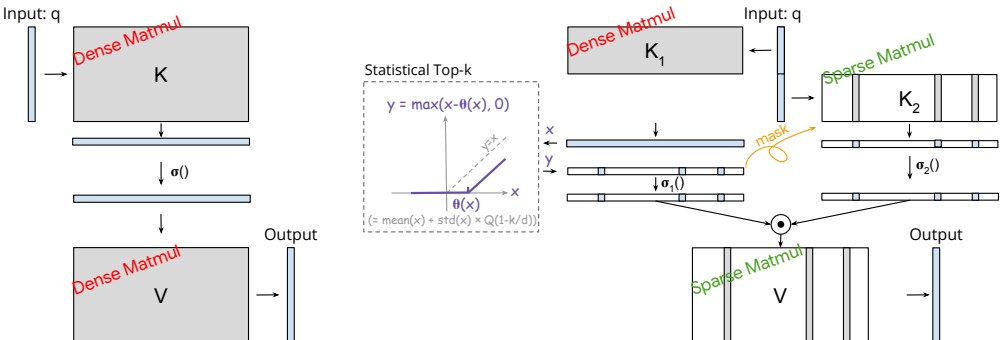

Figure 2: Architecture of Spark FFN and Spark Attention. **(Left)** Unified illustration of standard FFN (*i.e.,* eq. (1)) and standard Attention (*i.e.,* eq. (6)). In the case of FFN, $q \in \mathbb{R}^{d_{\text{model}}}$ is the input, $K$ and $V$ are the first and second layer weights, respectively, and $\sigma()$ is GELU. In the case of Attention, $q \in \mathbb{R}^{d_{\text{attn}}}$ is the query, $K$ and $V$ are key and value matrices, respectively, and $\sigma()$ is softmax. **(Right)** Unified illustration of Spark FFN (*i.e.*, eq. (2)) and Spark Attention (*i.e.*, eq. (7)). In the case of Spark FFN, $\sigma_1()$ is GELU and $\sigma_2()$ is identity. In the case of Spark Attention, $\sigma_1()$ is softmax and $\sigma_2()$ is softplus. In both cases, $\text{Statistical-Top}_k$ (*i.e.,* eq. (10)) is applied to introduce sparsity, which enables *sparse* matrix multiplication with $K_2$ and $V$ that reduces FLOPs count.

In order to reduce FLOPs count in the first layer of FFN as well, we introduce Spark FFN as follows:

$$\text{Spark-FFN}(q; K_1, K_2, V, k, r) \stackrel{\text{def}}{=} V \cdot \left( \sigma \left( \text{Top}_k(K_1^\top \cdot q[:r]) \right) \odot \left( K_2^\top \cdot q[r:] \right) \right). \qquad (2)$$

In above, $K_1 \subseteq \mathbb{R}^{r \times d_{\text{ff}}}$, $K_2 \subseteq \mathbb{R}^{(d_{\text{model}} - r) \times d_{\text{ff}}}$, and $V \subseteq \mathbb{R}^{d_{\text{model}} \times d_{\text{ff}}}$ are trainable parameters, and the activation $\sigma()$ is taken to be GELU [37] following Gemma. $\text{Top}_k$ is introduced for obtaining sparsity, with $k$ being a hyper-parameter specifying the sparsity level. Concretely, $\text{Top}_k$ preserves the largest $k$ values in the activation vector, while setting other values to 0. In this paper, we use the efficient *statistical* top-k presented in Section 3, which avoids sorting activation values. Finally, the input $q$ is split into $q[:r]$ and $q[r:]$, which contain the first $r$ and the rest of the dimensions, respectively, with $r$ being a hyper-parameter. It is introduced so that the term $K_1^\top q[:r]$ serves as a low-rank predictor of the location of the nonzero entries, which allows us to obtain efficiency benefits in computing $K_2^\top q[r:]$ and the multiplication with $V$. This is discussed in detail below.

**FLOPs per Token.** Direct implementation of the Spark-FFN without exploiting sparsity has the same number of FLOPs as the standard FFN in eq. (1), i.e.,

$$2r \cdot d_{\text{ff}} + 2(d_{\text{model}} - r) \cdot d_{\text{ff}} + 2d_{\text{model}} \cdot d_{\text{ff}} = 4d_{\text{model}} \times d_{\text{ff}}, \qquad (3)$$

where the three terms are from multiplication with $K_1$, $K_2$, and $V$, respectively. In Spark-FFN, one may first compute the term $K_1^\top q[:r]$ as a low-rank predictor. After passing its output through $\text{Top}_k$, which selects approximately the $k$ most important entries, followed by the activation function $\sigma()$, we obtain a sparse output. Importantly, after obtaining the sparse output there is no need to perform the full computation of the other two matrix multiplications in eq. (2), i.e., $K_2^\top q[r:]$ and the multiplication with $V$. Instead, one can perform a sparse matrix multiplication with a drastically reduced FLOPs count:

$$2r \cdot d_{\text{ff}} + 2(d_{\text{model}} - r) \cdot k + 2d_{\text{model}} \cdot k = 2(d_{\text{ff}} - k) \cdot r + 4d_{\text{model}} \cdot k, \qquad (4)$$

which is an increasing function of $r$. In other words, $r$ controls the computation cost. We provide ablation study in Section C.4 to show that the best model quality is obtained when $r \approx \frac{d_{\text{model}}}{2}$. In this case, the total FLOP count of Spark FFN is approximately $d_{\text{model}} \cdot d_{\text{ff}} + 3 \cdot d_{\text{model}} \cdot k$, which is a 4-times reduction from eq. (3) when $k$ is small.

**Relation to gated activation.** Many of the most recent Transformers, including Gemma, use a variant of the standard FFN in eq. (1) where the activation function is replaced with a gated one:

$$\text{Gated-FFN}(q; K_1, K_2, V) = V \cdot \left( \sigma(K_1^\top q) \odot (K_2^\top q) \right). \qquad (5)$$

In above, $\{K_1, K_2, V\} \subseteq \mathbb{R}^{d_{\text{model}} \times d'_{\text{ff}}}$. Note that when compared with the FFN in eq. (1) for quality studies, $d'_{\text{ff}}$ is usually taken to be $2/3 \cdot d_{\text{ff}}$ to be iso-parameter count [77].

Our Spark FFN in eq. (2) bears some resemblance to Gated FFN in that both have two linear maps in the first layer and one in the second layer. The difference lies in that 1) Spark FFN adds a $\text{Top}_k$ to obtain sparsity, and 2) the input to the first layers of Spark FFN are obtained from splitting the dimensions of the input.

## 2.2 Spark Attention

In a standard multi-head attention layer, an input $\boldsymbol{x} \in \mathbb{R}^{d_{\text{model}}}$ is mapped to a query, a key, and a value vector of dimension $d_{\text{attn}}$ as $\boldsymbol{q}^{(i)} = \boldsymbol{W}_Q^{(i)} \boldsymbol{x} \in \mathbb{R}^{d_{\text{attn}}}, \boldsymbol{k}^{(i)} = \boldsymbol{W}_K^{(i)} \boldsymbol{x} \in \mathbb{R}^{d_{\text{attn}}}, \boldsymbol{v}^{(i)} = \boldsymbol{W}_V^{(i)} \boldsymbol{x} \in \mathbb{R}^{d_{\text{attn}}}$ for each head $i$. Here, $\{\boldsymbol{W}_Q^{(i)}, \boldsymbol{W}_K^{(i)}, \boldsymbol{W}_V^{(i)}\} \subseteq \mathbb{R}^{d_{\text{attn}} \times d_{\text{model}}}$ are trainable weights.

Collecting all the key and value vectors in the context of $\boldsymbol{x}$ into $\boldsymbol{K}^{(i)} = [\boldsymbol{k}_1^{(i)}, \ldots, \boldsymbol{k}_{n_{\text{ctx}}}^{(i)}] \in \mathbb{R}^{d_{\text{attn}} \times n_{\text{ctx}}}$ and $\boldsymbol{V}^{(i)} = [\boldsymbol{v}_1^{(i)}, \ldots, \boldsymbol{v}_{n_{\text{ctx}}}^{(i)}] \in \mathbb{R}^{d_{\text{attn}} \times n_{\text{ctx}}}$, attention conducts the following computation:

$$\text{Standard-Attention}(\boldsymbol{q}; \boldsymbol{K}, \boldsymbol{V}) \stackrel{\text{def}}{=} \boldsymbol{V} \cdot \text{softmax}\left(\boldsymbol{K}^\top \boldsymbol{q}\right) \in \mathbb{R}^{d_{\text{attn}}}, \tag{6}$$

where we omit the dependency on $i$ for simplicity. The computation cost associated with eq. (6) is $4d_{\text{attn}} \cdot n_{\text{ctx}}$ for each head. Finally, output from all heads are concatenated followed by a linear map to project to $d_{\text{model}}$.

Note that eq. (6) has the same form as FFN in eq. (1) except for the choice of nonlinearity. Hence, following a similar strategy in obtaining Spark FFN, here we present Spark Attention as

$$\text{Spark-Attention}(\boldsymbol{q}; \boldsymbol{K}, \boldsymbol{V}, k, r) \stackrel{\text{def}}{=} \boldsymbol{V} \cdot \left(\sigma_1\left(\text{Top}_k^{(-\infty)}(\boldsymbol{K}_1^\top \boldsymbol{q}[:r])\right) \odot \sigma_2\left(\boldsymbol{K}_2^\top \boldsymbol{q}[r:]\right)\right). \tag{7}$$

In above, $\boldsymbol{K}_1 \in \mathbb{R}^{r \times n_{\text{ctx}}}$ and $\boldsymbol{K}_2 \in \mathbb{R}^{(d_{\text{attn}} - r) \times n_{\text{ctx}}}$ contain the first $r$ rows and the rest of the rows from $\boldsymbol{K}$, respectively, and $\sigma_1$ and $\sigma_2$ are nonlinear functions. Empirically, we find that taking $\sigma_1 = \text{softmax}$ and $\sigma_2 = \text{softplus}$ gives good results. Finally, $\text{Top}_k^{(-\infty)}$ refers to an operator that keeps the largest $k$ values of the input while setting the rest to $-\infty$. This operator be implemented using *statistical* top-k, which we explain in Section 3.

**FLOPs per Token.** With a direct implementation the number of FLOPs in eq. (7) is given by $4d_{\text{attn}} \cdot n_{\text{ctx}}$, which is the same as the FLOPs for eq. (6). However, by noting that the output of the softmax is expected to be sparse with approximately $k$ nonzero entries, the computation costs associated with $\boldsymbol{K}_1^\top \boldsymbol{q}[:r]$ and in the multiplication with $\boldsymbol{V}$ can be drastically reduced. In particular, if we take $r = \frac{d_{\text{attn}}}{2}$ then the FLOPs per token becomes

$$d_{\text{model}} n_{\text{ctx}} + 3d_{\text{model}} \min\{k_{\text{attn}}, n_{\text{ctx}}\}, \tag{8}$$

which is nearly a $4\times$ reduction when $k_{\text{attn}} \ll n_{\text{ctx}}$.

## 3 Statistical Top-k

This section introduces $\text{Statistical-Top}_k$, an efficient algorithm for implementing the $\text{Top}_k$ operators in Spark FFN (i.e. eq. (2)) and Spark Attention (i.e. eq. (7)).

Recall that the *soft-thresholding operator* is defined for an arbitrary vector $\boldsymbol{x} \in \mathbb{R}^d$ and a scalar threshold $\theta \in \mathbb{R}$ as

$$\text{Soft-Threshold}(\boldsymbol{x}, \theta) \stackrel{\text{def}}{=} \max\{\boldsymbol{x} - \theta \cdot \mathbf{1}, \mathbf{0}\} \in \mathbb{R}^d, \tag{9}$$

where $\mathbf{1}$ and $\mathbf{0}$ are $d$-dimensional vectors with all entries equal to 1 and 0, respectively. The soft-thresholding operator shifts each entry of $\boldsymbol{x}$ to the left by $\theta$ and then thresholds the result at zero.

We define $\text{Statistical-Top}_k$ as the following mapping from $\mathbb{R}^d$ to $\mathbb{R}^d$:

$$\text{Statistical-Top}_k(\boldsymbol{x}) \stackrel{\text{def}}{=} \text{Soft-Threshold}(\boldsymbol{x}, \theta(\boldsymbol{x}, k)), \tag{10}$$

where

$$\theta(\boldsymbol{x}, k) \stackrel{\text{def}}{=} \text{mean}(\boldsymbol{x}) + \text{std}(\boldsymbol{x}) \cdot Q(1 - \frac{k}{d}). \tag{11}$$

In above, we define $\text{mean}(\boldsymbol{x}) \stackrel{\text{def}}{=} \frac{1}{d} \sum_{i=1}^d x_i$ and $\text{std}(\boldsymbol{x}) \stackrel{\text{def}}{=} \sqrt{\frac{1}{d-1} \sum_{i=1}^d (x_i - \text{mean}(\boldsymbol{x}))^2}$, which compute the sample mean and standard deviation of the entries of the input $\boldsymbol{x}$, respectively. $Q(\cdot)$ is

the quantile function (i.e., inverse of the cumulative distribution function) of the standard Gaussian distribution. In Spark Transformer, eq. (10) is used as the $\mathrm{Top}_k$ operator in eq. (2). For the operator $\mathrm{Top}_k^{(-\infty)}$ in eq. (7), a slight variant of eq. (10) is used where the entries below the threshold $\theta(\boldsymbol{x}, k)$ are set to $-\infty$ instead of $0$.

$\mathrm{Statistical}\text{-}\mathrm{Top}_k$ operates by first computing a threshold $\theta(\boldsymbol{x}, k)$ such that approximately $k$ entries of $\boldsymbol{x}$ exceed it, and then applying the soft-thresholding operator with this threshold to $\boldsymbol{x}$ to obtain a sparse output. We discuss these two components in the next two subsections.

### 3.1 Threshold Estimation

The threshold $\theta(\boldsymbol{x}, k)$ in eq. (10) is designed such that, if the entries of $\boldsymbol{x}$ are drawn from a Gaussian distribution, approximately $k$ out of the $d$ entries will exceed this threshold. To understand this, let $\mu$ and $\sigma$ denote the mean and standard deviation of the underlying Gaussian distribution. Its quantile function is given by $\mu + \sigma \cdot Q(p)$ for $p \in (0, 1)$. Consequently, due to the properties of quantile functions, we expect roughly $p \cdot d$ entries of $\boldsymbol{x}$ to exceed $\mu + \sigma \cdot Q(1 - p)$. In practice, since $\mu$ and $\sigma$ are unknown, they are replaced with the sample mean $\mathrm{mean}(\boldsymbol{x})$ and the sample standard deviation $\mathrm{std}(\boldsymbol{x})$, respectively.

The following theorem formalizes this argument.

**Theorem 3.1.** *Let $\boldsymbol{x} \in \mathbb{R}^d$ be a vector with entries drawn i.i.d. from $\mathcal{N}(\mu, \sigma^2)$. For any $1 \leq k \leq d-1$, let $\theta(\boldsymbol{x}, k)$ be a scalar defined in eq. (10). Take any $\delta \in (0, 1)$ and assume $d \geq \max\{2, \log \frac{6}{\delta}\}$. With a probability of at least $1 - \delta$, the number of entries of $\boldsymbol{x}$ that are greater than $\theta(\boldsymbol{x}, k)$, i.e., $\mathbf{card}\left(\{i \in [d] \mid x_i > \theta(\boldsymbol{x}, k)\}\right)$, satisfies*

$$\frac{|\mathbf{card}\left(\{i \in [d] \mid x_i > \theta(\boldsymbol{x}, k)\}\right) - k|}{d} \leq 4 \sqrt{\frac{\log \frac{6}{\delta}}{d}} \left(1 + \sqrt{-2 \log \min\left\{\frac{k}{d}, 1 - \frac{k}{d}\right\}}\right). \quad (12)$$

Theorem 3.1 provides a relative error bound between $k$ and the true number of entries of $\boldsymbol{x}$ that exceed $k$. This bound is maximized when $k = 1$ or $k = d - 1$. Consequently, the worst-case bound is $O\left(\sqrt{\frac{\log d \cdot \log \frac{1}{\delta}}{d}}\right)$ which vanishes as $d$ increases. Notably, the error bound becomes $O\left(\sqrt{\frac{\log \frac{1}{\delta}}{d}}\right)$ when $k = \Theta(d)$, demonstrating even faster convergence.

**Computation cost.** The computation of the threshold $\theta(\mathbf{x}, k)$ is highly efficient and resembles the operations used in LayerNorm layers, requiring only $2d$ FLOPs to compute the mean and standard deviation of the samples. This contrasts sharply with a naive sorting-based approach, which has $O(d \log d)$ complexity.

While the Gaussian quantile function $Q(\cdot)$ lacks a closed-form solution, high-precision piecewise approximation algorithms with constant complexity are available in standard software packages like SciPy [86], readily applicable to our needs.

### 3.2 Sparsification

Given the threshold $\theta(\boldsymbol{x}, k)$, a straightforward approach to obtain a sparse vector is to set all entries of $\boldsymbol{x}$ below the threshold to zero, preserving the remaining values. This operator, sometimes referred to as *hard thresholding* [10], suffers from discontinuity, potentially hindering its suitability for gradient-descent-based training.

To address this, $\mathrm{Statistical}\text{-}\mathrm{Top}_k$ employs the soft-thresholding operator defined in eq. (9) [8]. This operator first shrinks all entries of $\boldsymbol{x}$ by the threshold $\theta(\boldsymbol{x}, k)$ and then sets all entries below $0$ to $0$. Soft thresholding offers the advantages of being continuous and differentiable almost everywhere (except when entries of $\boldsymbol{x}$ coincide with $\theta(\boldsymbol{x}, k)$).

For complete differentiability, one can utilize a smoothing function like the Huber loss [39], defined element-wise on an input $\boldsymbol{x}$ as:

$$\mathrm{Huber}(x; \delta) \stackrel{\text{def}}{=} \begin{cases} \frac{1}{2} x^2 & \text{for } |x| < \delta, \\ \delta \cdot (|x| - \frac{1}{2}\delta) & \text{otherwise.} \end{cases} \quad (13)$$

The continuous differentiability of the mapping $\boldsymbol{x} \mapsto \mathrm{Huber}(\mathrm{Statistical}\text{-}\mathrm{Top}_k(\boldsymbol{x}); \delta)/\delta$ is established below:

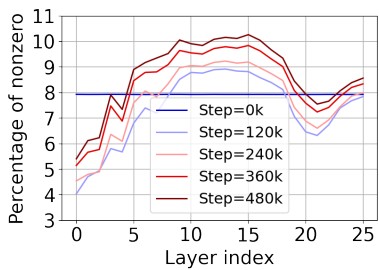

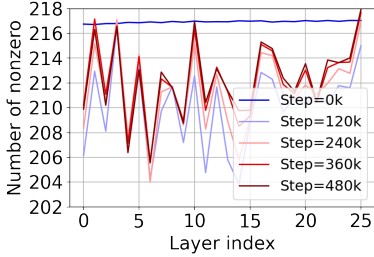

(a) Sparsity level in Spark FFN    (b) Sparsity level in Spark Attention

Figure 3: Sparsity in the intermediate activation of Spark FFN and Spark Attention across 26 layers at selected training steps. For FFN we report the percentage of nonzero entries out of $d_{\text{ff}} = 13824$ entries. For Attention, we report the number of nonzero entries (i.e., attended tokens). Our hyper-parameter choice is to have 8% nonzeros in Spark FFN and at most 256 nonzeros in Spark Attention.

**Theorem 3.2.** *For any $\delta > 0$, the function $\mathbb{R}^d \to \mathbb{R}^d$ defined as*

$$\text{Huber}(\text{Statistical-Top}_k(\boldsymbol{x}); \delta) \, / \, \delta \tag{14}$$

*is continuously differentiable.*

Note that eq. (14) converges to $\text{Statistical-Top}_k(\boldsymbol{x})$ as $\delta \to 0$, which can be seen as $\text{Huber}(x; \delta)/\delta \to |x|$ and $\text{Statistical-Top}_k(\boldsymbol{x})$ is always non-negative. In practice, however, we find that using a non-zero $\delta$ does not improve model quality, and therefore we set $\delta = 0$ for simplicity.

Finally, soft thresholding admits a variational form (see, e.g., [67]):

$$\text{Soft-Threshold}(\boldsymbol{x}, \theta) = \arg\min_{\boldsymbol{z} \geq \boldsymbol{0}} \theta \|\boldsymbol{z}\|_1 + \frac{1}{2}\|\boldsymbol{x} - \boldsymbol{z}\|_2^2. \tag{15}$$

This formulation seeks a vector $\mathbf{z}$ that minimizes both its squared $\ell_2$ distance to the input $\boldsymbol{x}$ and its $\ell_1$ norm, with the threshold $\theta$ balancing these terms. Given the sparsity-promoting nature of the $\ell_1$ norm, soft thresholding effectively finds a sparse approximation of the input $\boldsymbol{x}$. This variational form also connects $\text{Statistical-Top}_k$ with other top-$k$ algorithms in the literature; see Section D.3.

## 4   Experiments

In this section, we present an experimental evaluation of Spark Transformer using the Gemma-2 2B model. Gemma-2 2B is a decoder-only Transformer with 2 billion parameters, pretrained on 2 trillion tokens of primarily English text data (see [30] for details). To evaluate Spark Transformer, we train a model by substituting the standard FFN and Attention in Gemma-2 2B with their Spark Transformer counterparts (Spark FFN and Spark Attention, respectively). This Spark Transformer model is trained using the same procedure and data as the Gemma-2 2B model.

**Implementation details.** Gemma-2 uses a model dimension of $d_{\text{model}} = 2304$. **For FFN**, Gemma-2 uses the Gated FFN in eq. (5) with $d'_{\text{ff}} = 9216$. We replace it with Spark FFN in eq. (2) with $d_{\text{ff}} = 13824$ so that the parameter count keeps the same. In addition, we take $k$ to be 1106, which gives a sparsity level of 8%, and $r = 1024 \approx d_{\text{model}}/2$ (due to sharding constraints, $r$ can only be a multiple of 256). **For Attention**, Gemma-2 alternates between a global attention that have a span of 8192 tokens, and a local attention with a 4096 window size, both with $d_{\text{attn}} = 256$. We replace both with Spark Attention in eq. (6) where for the latter we use the same 4096 window size. For hyper-parameters, we use $k = 256$, i.e. each token attends to at most 256 tokens, and $r = 128 = d_{\text{attn}}/2$. Gemma-2 uses Rotary Position Embedding [84] which is applied to $\boldsymbol{q}$ and the columns of $\boldsymbol{K}$ in eq. (6). For Spark Attention in eq. (7), we apply this position encoding to $\boldsymbol{q}[:r]$, $\boldsymbol{q}[r:]$, the columns of $\boldsymbol{K}_1$, and the columns of $\boldsymbol{K}_2$.

### 4.1   Quality

We evaluate Spark Transformer on a suite of benchmarks that are used in the Gemma-2 paper [30], and report the result in Figure 1b. We observe that Spark Transformer matches the quality of Gemma-2 while having a drastically reduced FLOP count per token.

The near-quality neutrality of Spark Transformer distinguishes it from related work on enforcing activation sparsity in FFN, which often lead to a quality loss. To demonstrate this, we pretrain variants

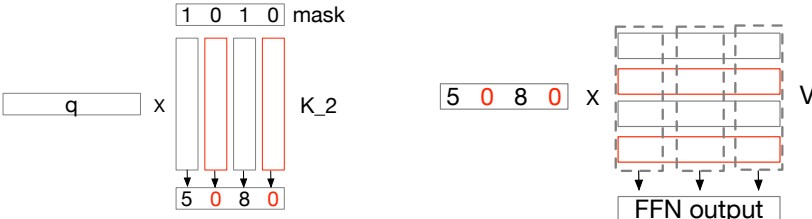

Red: skipped memory access /compute

(a) Vector-Masked Matrix Multiplication  (b) Sparse Vector-Matrix Multiplication

Figure 4: Illustration of the matrix multiplication implementation using sparse activation. (a) Vector-Masked Matrix Multiplication takes a dense vector $q[r:]$, a dense matrix $K_2^\top$, and a mask from statistical top-$k$ on $K_1^\top q[:r]$ to compute $u := (K_2^\top q[r:]) \odot$ mask. It skips memory loading and compute associated with the masked columns. (b) Sparse Vector-Matrix Multiplication takes a sparse activation vector $u$ to compute weighted sum of rows in the dense matrix $V$. It skips loading and computation of rows corresponding to 0's in $u$. To optimize performance, we implement Sparse Vector-Matrix Multiplication using tiling, which helps minimize cross-CPU core synchronization.

of Gemma-2 2B with 1) activation function switched to ReLU [92], 2) activation function switched to ReLU$^2$ [64], and 3) Top-k thresholding applied before GELU. Due to high training cost, all models are pretrained for 1/6 of the standard Gemma-2 training iterations. In Figure 1a, we report the training losses and FLOPs per token relative to those of the standard Gemma-2. It can be seen that these models either suffer from a large quality loss (i.e., ReLU and Topk), or do not lead to a sufficient FLOPs reduction (i.e., ReLU$^2$). In contrast, our Spark FFN achieves less quality loss with more FLOPs reduction.[6] Finally, the combination of Spark FFN with Spark Attention introduces additional FLOPs reduction and quality benefits, enabling an almost neutral quality of Spark Transformer with a large overall FLOPs reduction.

### 4.2 Sparsity

To verify the effectiveness of statistical top-$k$, we report the level of sparsity measured in terms of percentage of nonzeros in FFN and the number of nonzeros in Attention. At the beginning of model training, we observe that statistical top-$k$ produces close to 8% nonzeros in FFN (see Figure 3a), which aligns well with our hyper-parameter choice of using $k/d_{\mathrm{ff}} = 8\%$ in Spark FFN. This is expected as the model parameters, particularly $K$ in Spark FFN, are randomly initialized, hence the entries of the activation maps are drawn from a Gaussian distribution which is in accordance with the assumption of statistical top-$k$. The Gaussian assumption is no longer guaranteed after training, but we empirically observe it to hold approximately (see Section C.1) and statistical top-$k$ reliably produce a sparsity level close to 8% until the end of training at $480k$ steps. Sparsity in attention is reported in Figure 3b, which show that the number of attended tokens is below our hyper-parameter choice of 256 in Spark Attention throughout training. In particular, the numbers are much smaller because the results are from averaging over all tokens many of which have a context length of less than 256. Finally, we observe comparable levels of sparsity during evaluation (see Section C.2).

### 4.3 Inference Efficiency

We evaluate the efficiency benefits of Spark Transformer over standard Gemma-2 on both CPUs and GPUs. For CPU evaluation, we use gemma.cpp [33], the official C++ inference engine optimized for CPUs. For GPU evaluation, we use llama.cpp [31], a widely-used LLM inference engine which supports running a wide selection of LLM models on GPUs. We modify both frameworks to support sparse matrix multiplication operators, which exploit sparsity in both FFN and Attention layers. Our implementation leverages vector SIMD operations [80] existing in modern CPU, and customized CUDA kernel for GPU. In particular, our implementation reduces not only computational FLOPs but also memory bandwidth requirements, directly accelerating memory-bound scenarios; see Figure 4 for

---

[6]Notably, the only difference between Spark FFN and Topk, when both have 8% activated neurons, is in the introduction of the low-cost predictor. Hence, this predictor not only leads to FLOPs reduction but also (surprisingly?) improves model quality.

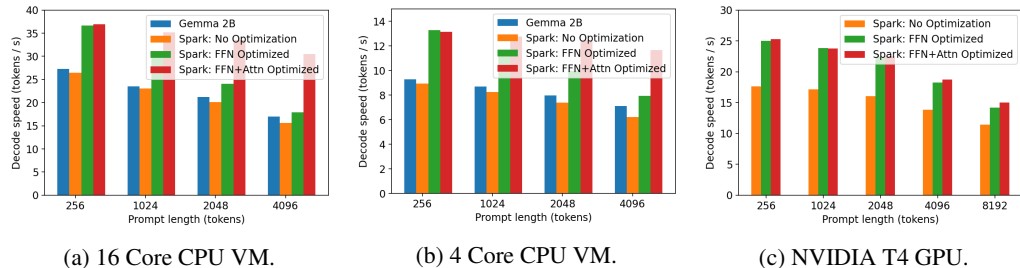

|  (a) 16 Core CPU VM. | (b) 4 Core CPU VM. | (c) NVIDIA T4 GPU. |

Figure 5: Spark Transformer decoding speedup from activation sparsity on various hardware platforms. We report decoding speed of Spark Transformer without hardware optimization for sparse activation, with hardware optimization for sparsity in Spark FFN only, and with hardware optimization for sparsity in both Spark FFN and Spark Attention. All experiments use a decode batch size of 1.

an illustration and Section B in Appendix for details. We show that Spark Transformer significantly improves the efficiency of transformer models, even in highly FLOP-constrained environments such as CPUs.

**CPU results.** Figure 5a and 5b report the decoding speed under varying prompt lengths on a 4-Core or a 16-Core CPU. We see that with hardware optimization for sparse activation in both Spark FFN and Spark Attention, a speedup that ranges from 1.35x to 1.79x can be achieved on a 16-Core CPU depending on the prompt length. For short prompts (e.g., 256 tokens) and long prompts (e.g. 4096 tokens), the hardware optimization for Spark FFN and Spark Attention provide the most speedup, respectively.

Figure 1c further highlights the efficiency of Spark Transformer in both prefill and decode phases. During the prefill, the prompt is usually chunked into batches since the process is bounded by memory bandwidth. This may reduce the benefit of activation sparsity as different tokens in a chunk may activate different subsets of parameters (in FFN) and attend to different subsets of tokens (in Attention). However, Figure 1c shows that Spark Transformer maintains strong performance with a chunk size of 64 tokens, following the default setup in `gemma.cpp`. A more detailed performance analysis of batching/chunking is provided in Section C.3. In addition, Spark Transformer significantly outperforms Gemma-2 during decoding (with batch size=1).

**GPU results.** We also evaluate the efficiency gain from sparsity on low-profile GPUs. Figure 5c reports the decoding speed under varying prompt lengths on an NVIDIA T4 GPU. Similar to the CPU case, we see Spark Transformer achieves decode speedup ranging from $1.25\times$ to $1.40\times$.

## 4.4 Training Efficiency

The introduction of a top-$k$ operator is expected to lead to a training slowdown due to the extra computational operators. This slowdown can be prohibitively large if one uses the standard approximate top-$k$ operator provided in JAX, namely `jax.lax.approx_max_k` [14]. Specifically, this JAX top-$k$ operator is optimized to achieve TPU peak performance and has a controllable recall target, which we vary on the x-axis in Figure 6. It can be observed that the JAX top-$k$ leads to more than $10\times$ slowdown even when operating on a small recall of 50%. In contrast, the slowdown from statistical top-$k$ is with a very small amount, demonstrating its efficiency. Finally, we do not provide the quality of models trained with JAX top-$k$ since such models take a very long time to train.

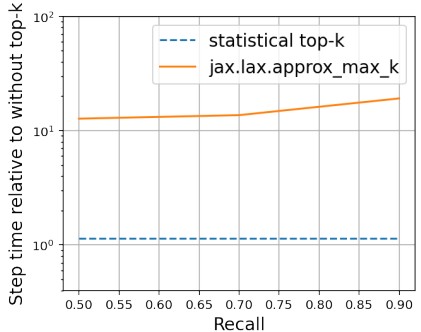

Figure 6: Comparison of training slowdown from using our statistical top-$k$ vs the standard top-$k$ (i.e., `jax.lax.approx_max_k` [14]) relative to not using any top-$k$.

## 5  Discussions

This paper introduces the Spark Transformer architecture to reduce the FLOPs in both the FFN and Attention of Transformer models. Because the FFN and Attention components dominate the

computational cost in large Transformers with long contexts, the *overall* FLOP count for decoding a token is also drastically reduced (see Table C.1), leading to notable computation efficiency benefits on appropriate hardwares. Specifically, the benefit is obtained by selectively activating only part of the model parameters and limiting the attended context for each input. This principle of sparse activation finds a compelling parallel in neuroscience, where studies reveal sparse activity patterns in the brain as a key factor in its remarkable efficiency [5, 7, 48]. While hardware limitations currently hinder the full exploitation of sparse activation, our demonstration of practical wall-time reduction of Spark Transformer on CPU and GPU, together with prior evidence for related techniques [58, 82] including TPU [89], highlight its potential. We hope that this work opens avenues for research into alternative hardware better suited for sparse computations, circumventing the hardware lottery [38] and potentially leading to greater efficiency gains in the future. Additional discussions on related work for activation sparsity, including Mixture-of-Experts, are provided in Section D.1.

This work also opens promising avenues for future research on efficient inference of large language models, particularly in combining Spark Transformer's architectural efficiency with other leading optimization techniques. We briefly discuss two key synergies here.

**Synergy with Speculative Decoding.** Spark Transformer is highly complementary to speculative decoding. As a target model, its faster inference directly accelerates the primary verification bottleneck. As a draft model, its near-quality neutrality and high speed make it an ideal candidate for generating high-quality drafts, potentially leading to higher token acceptance rates and greater overall speedups. A full discussion is provided in Section D.5.

**Synergy with Quantization.** We also hypothesize a strong synergy with quantization. The benefits are expected to be multiplicative, as Spark Transformer reduces the number of operations while quantization reduces their cost. More importantly, unlike standard pruning which preserves high-magnitude outliers, our statistical top-$k$ operator uses soft-thresholding. This shrinks the dynamic range of activation distribution, which may reduce sensitivity to activation quantization. A detailed analysis of this mechanism is available in Appendix section D.6.

## Acknowledgments and Disclosure of Funding

The authors acknowledge helpful discussion with Zonglin Li (Anthropic), Daliang Li (Anthropic), Amir Yazdanbakhsh (Google), Zeid Samoail (Google), Abhishek Kumar (Amazon), Vlad Feinberg (Google), Veeru Sadhanala (Google) at various stages of this project.

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

# Appendix A  Proofs

## A.1  Proof to Theorem 3.1

*Proof.* In this proof we write $\bar{x} \overset{\text{def}}{=} \text{mean}(x)$ and $s \overset{\text{def}}{=} \text{std}(x)$ for brevity.

We first establish the concentration bounds that the empirical mean and standard deviation, i.e., $\bar{x}$ and $s$ are close to the true mean and true standard deviation, i.e., $\mu$ and $\sigma$ of the underlying Gaussian, respectively. Recall from the definition of the chi-squared distribution that $(d-1)\frac{s^2}{\sigma^2} \sim \chi^2(d-1)$. Using the Laurent-Massart bound on the tail probability of the chi-squared distribution [47, Corollary of Lemma 1], we have

$$\Pr\left(\left|(d-1)\frac{s^2}{\sigma^2} - (d-1)\right| \geq 2\sqrt{(d-1)t} + 2t\right) \leq 2e^{-t}$$

for every $t > 0$. We set $t = \log \frac{6}{\delta}$. Then, with a probability of at least $1 - \delta/3$, we have

$$(d-1)\left|\frac{s^2}{\sigma^2} - 1\right| < 2\sqrt{(d-1)\log\frac{6}{\delta}} + 2\log\frac{6}{\delta},$$

which implies

$$\left|\frac{s^2}{\sigma^2} - 1\right| < 2\sqrt{\frac{\log\frac{6}{\delta}}{d-1}} + 2\frac{\log\frac{6}{\delta}}{d-1} \leq 4\sqrt{\frac{\log\frac{6}{\delta}}{d}} + 4\frac{\log\frac{6}{\delta}}{d} \leq 8\sqrt{\frac{\log\frac{6}{\delta}}{d}},$$

where the last inequality uses the assumption that $d \geq \max\{2, \log\frac{6}{\delta}\}$. By rearranging the terms, we get

$$\sigma\left(1 - 8\sqrt{\frac{\log\frac{6}{\delta}}{d}}\right) \leq \sigma\sqrt{\max\left\{1 - 8\sqrt{\frac{\log\frac{6}{\delta}}{d}}, 0\right\}} \leq s \leq \sigma\sqrt{1 + 8\sqrt{\frac{\log\frac{6}{\delta}}{d}}} \leq \sigma\left(1 + 8\sqrt{\frac{\log\frac{6}{\delta}}{d}}\right),$$

which simplifies to

$$|s - \sigma| \leq 8\sigma\sqrt{\frac{\log\frac{6}{\delta}}{d}}. \tag{A.1}$$

eq. (A.1) provides a concentration bound for $s$. We now proceed to deriving a bound for $\mu$. Towards that, notice that $\frac{\bar{x}-\mu}{\sigma/\sqrt{d}} \sim \mathcal{N}(0,1)$. By using the Mill's inequality that upper bounds the tail probability of a standard normal distribution (i.e., if $Z \sim \mathcal{N}(0,1)$ and $t > 0$, then $\Pr(|Z| > t) \leq \frac{e^{-t^2/2}}{t}$), we have

$$\Pr\left(\left|\frac{\bar{x}-\mu}{\sigma/\sqrt{d}}\right| > \sqrt{2\log\frac{3}{\delta}}\right) \leq \frac{\delta/3}{\sqrt{2\log\frac{3}{\delta}}} \leq \delta/3.$$

Therefore, with probability at least $1 - \delta/3$, we have

$$\left|\frac{\bar{x}-\mu}{\sigma/\sqrt{d}}\right| \leq \sqrt{2\log\frac{3}{\delta}},$$

which yields

$$|\bar{x} - \mu| \leq \sigma\sqrt{\frac{2\log\frac{3}{\delta}}{d}}. \tag{A.2}$$

Combining eq. (A.1) and eq. (A.2), with probability at least $1 - 2\delta/3$, we have

$$\left| \theta(\boldsymbol{x}, k) - (\mu + \sigma Q(1 - \frac{k}{d})) \right| \tag{A.3}$$

$$\leq |\bar{x} - \mu| + |s - \sigma| \left| Q(1 - \frac{k}{d}) \right| \tag{A.4}$$

$$\leq \sigma \sqrt{\frac{2 \log \frac{3}{\delta}}{d}} + 8\sigma \sqrt{\frac{\log \frac{6}{\delta}}{d}} \left| Q(1 - \frac{k}{d}) \right| . \tag{A.5}$$

We define the empirical cumulative distribution function (ECDF) of $x_1, x_2, \ldots, x_d$ as

$$\hat{F}_d(x) = \frac{1}{d} \sum_{i \in [d]} \mathbf{1}_{\{x_i \leq x\}}.$$

Then, the number of the entries of $\boldsymbol{x}$ that are greater than $\theta(\boldsymbol{x}, k)$ may be written as

$$\mathbf{card}\left(\{i \in [d] \mid x_i > \theta(\boldsymbol{x}, k)\}\right) = \sum_{i \in [d]} \mathbf{1}_{\{x_i > \theta(\boldsymbol{x}, k)\}} = d\left(1 - \hat{F}_d(\theta(\boldsymbol{x}, k))\right).$$

Let $F$ denote the cumulative distribution function (CDF) of $\mathcal{N}(\mu, \sigma^2)$. By the Dvoretzky-Kiefer-Wolfowitz inequality [28, 63], we have

$$\Pr\left(\sup_{u \in \mathbb{R}} \left| \hat{F}_d(u) - F(u) \right| > t\right) \leq 2e^{-2dt^2} .$$

Taking $t = \sqrt{\frac{1}{2d} \log \frac{6}{\delta}}$ and $u = \theta(\boldsymbol{x}, k)$, we obtain

$$\Pr\left(\left| \hat{F}_d(\theta(\boldsymbol{x}, k)) - F(\theta(\boldsymbol{x}, k)) \right| > \sqrt{\frac{1}{2d} \log \frac{6}{\delta}}\right) \leq \frac{\delta}{3}. \tag{A.6}$$

Applying the union bound on eq. (A.3) and eq. (A.6), we obtain that the following holds with probability at least $1 - \delta$:

$$\left| \hat{F}_d(\theta(\boldsymbol{x}, k)) - (1 - \frac{k}{d}) \right| \tag{A.7}$$

$$= \left| \hat{F}_d(\theta(\boldsymbol{x}, k)) - F(\mu + \sigma Q(1 - \frac{k}{d})) \right| \tag{A.8}$$

$$\leq \left| \hat{F}_d(\theta(\boldsymbol{x}, k)) - F(\theta(\boldsymbol{x}, k)) \right| + \left| F(\theta(\boldsymbol{x}, k)) - F(\mu + \sigma Q(1 - \frac{k}{d})) \right| \tag{A.9}$$

$$\leq \sqrt{\frac{1}{2d} \log \frac{6}{\delta}} + \frac{1}{\sqrt{2\pi}\sigma} \left| \theta(\boldsymbol{x}, k) - (\mu + \sigma Q(1 - \frac{k}{d})) \right| \tag{A.10}$$

$$\leq \sqrt{\frac{1}{2d} \log \frac{6}{\delta}} + \frac{1}{\sqrt{2\pi}} \left( \sqrt{\frac{2 \log \frac{3}{\delta}}{d}} + 8\sqrt{\frac{\log \frac{6}{\delta}}{d}} \left| Q(1 - \frac{k}{d}) \right| \right) \tag{A.11}$$

$$\leq 4\sqrt{\frac{\log \frac{6}{\delta}}{d}} \left( 1 + \left| Q(1 - \frac{k}{d}) \right| \right) . \tag{A.12}$$

In the above expression, the first equality stems directly from the definitions of $F(\cdot)$ and $Q(\cdot)$, which gives

$$F(\theta(\boldsymbol{x}, k)) = F(\mu + \sigma Q(1 - \frac{k}{d})) = \Phi(Q(1 - \frac{k}{d})) = 1 - \frac{k}{d},$$

where $\Phi$ denotes the CDF of the standard normal distribution.

To simplify eq. (A.7), we consider two cases:

- If $k \leq d/2$, by Mill's inequality, we have

$$1 - \Phi(\sqrt{2 \log \frac{d}{k}}) \leq \frac{e^{-(\sqrt{2 \log \frac{d}{k}})^2/2}}{\sqrt{2 \log \frac{d}{k}}} = \frac{e^{-(\sqrt{2 \log \frac{d}{k}})^2/2}}{\sqrt{2 \log \frac{d}{k}}} = \frac{k/d}{\sqrt{2 \log \frac{d}{k}}} \leq \frac{k}{d},$$

where the last inequality is because $2 \log \frac{d}{k} \geq 1$. Therefore

$$1 - \frac{k}{d} \leq \Phi(\sqrt{2 \log \frac{d}{k}}),$$

which gives

$$Q(1 - \frac{k}{d}) \leq \sqrt{2 \log \frac{d}{k}} = \sqrt{-2 \log \frac{k}{d}}.$$

- If $k > d/2$, we have

$$\left| Q(1 - \frac{k}{d}) \right| = \left| Q(1 - \frac{d-k}{d}) \right| \leq \sqrt{-2 \log \frac{d-k}{d}}.$$

Combining the two cases, we get

$$\left| Q(1 - \frac{k}{d}) \right| \leq \sqrt{-2 \log \min \left\{ \frac{k}{d}, 1 - \frac{k}{d} \right\}}.$$

Plugging this into eq. (A.7), we obtain

$$\left| \hat{F}_d(\theta(\boldsymbol{x}, k)) - (1 - \frac{k}{d}) \right| \leq 4 \sqrt{\frac{\log \frac{6}{\delta}}{d}} \left( 1 + \sqrt{-2 \log \min \left\{ \frac{k}{d}, 1 - \frac{k}{d} \right\}} \right).$$

Recall $\mathbf{card}\left(\{i \in [d] \mid x_i > \theta(\boldsymbol{x}, k)\}\right) = d\left(1 - \hat{F}_d(\theta(\boldsymbol{x}, k))\right)$. We conclude that with probability at least $1 - \delta$, we have

$$\left| \mathbf{card}\left(\{i \in [d] \mid x_i > \theta(\boldsymbol{x}, k)\}\right) - k \right| \leq 4 \sqrt{d \log \frac{6}{\delta}} \left( 1 + \sqrt{-2 \log \min \left\{ \frac{k}{d}, 1 - \frac{k}{d} \right\}} \right).$$

$\square$

## A.2  Proof to Theorem 3.2

*Proof.* The Huber statistical top-$k$ in eq. (14) may be written as

$$\text{Huber}(\text{Statistical-Top}_k(\boldsymbol{x}); \delta)/\delta = \text{Huber}(\text{Soft-Threshold}(\boldsymbol{x}, \theta(\boldsymbol{x}, k)))/\delta, \tag{A.13}$$

where $\theta(\boldsymbol{x}, k)$ is defined in eq. (10). This function is the (multivariate) composition of two functions, namely, $\theta = \theta(\boldsymbol{x}, k)$ and $\text{Huber}(\text{Soft-Threshold}(\boldsymbol{x}, \theta))$. In particular, the former is continuously differentiable (i.e., $C^1$) in $\boldsymbol{x}$, since it is simply a linear combination of sample mean and sample standard deviation both of which are $C^1$ functions. To establish the theorem, we only need to show that $\text{Huber}(\text{Soft-Threshold}(\boldsymbol{x}, \theta))$ is also a $C^1$ function in $(\boldsymbol{x}, \theta)$.

By definition, $\text{Huber}(\text{Soft-Threshold}(\boldsymbol{x}, \theta))$ is defined entry-wise on $\boldsymbol{x}$ as

$$\text{Huber}(\text{Soft-Threshold}(x, \theta)) = \begin{cases} \delta x - \delta \theta - \frac{1}{2}\delta, & \text{if } x > \theta + \delta; \\ \frac{1}{2}(x - \theta)^2, & \text{if } \theta \leq x \leq \theta + \delta; \\ 0, & \text{if } x < \theta. \end{cases} \tag{A.14}$$

From here it is easy to check that $\text{Huber}(\text{Soft-Threshold}(x, \theta))$ is continuous in $(x, \theta)$. Its gradient with respect to $(x, \theta)$ is given by

$$\frac{\partial \text{Huber}(\text{Soft-Threshold}(x, \theta))}{\partial (x, \theta)} = \begin{cases} (\delta, -\delta), & \text{if } x > \theta + \delta; \\ (x - \theta, \theta - x) & \text{if } \theta \leq x \leq \theta + \delta; \\ (0, 0), & \text{if } x < \theta, \end{cases} \tag{A.15}$$

which is also continuous. This concludes the proof. $\square$

## Appendix B    Implementation Details on Sparse Matrix Multiplications

We describe how we implement sparse matrix multiplications for Spark FFN and Attention in `gemma.cpp` for CPU and `llama.cpp` for GPU. We start by focusing on a batch size of one for decoding before expanding our discussion to larger batch sizes and prefill.

With batch size of 1, both Spark FFN and Spark Attention utilize two types of sparse vector-matrix multiplication: vector-masked matrix multiplication and sparse vector-matrix multiplication (Figure 4). Given a vector $q$ and a matrix $w$, vector-masked matrix multiplication multiplies $q$ with the non-masked columns of $w$ based on a masking vector $m$. Masked columns yield a zero output. Sparse vector-matrix multiplication, on the other hand, involves a vector that contains many zeroes being multiplied by a dense matrix.

In Spark FFN, we perform vector-masked matrix multiplication for $\boldsymbol{K}_2^\top \boldsymbol{q}[r:]$ (Figure 4a). The masking vector is generated from the output of $\text{Statistical-Top}_k(\boldsymbol{K}_1^\top \boldsymbol{q}[:r])$. Based on the mask, Spark FFN skips loading the masked columns of $w$ from DRAM (in CPU setup) or HBM (in GPU setup) and the associated computations. On CPU, Spark FFN utilizes SIMD operations (as in the original Gemma implementation). To further enhance performance, Spark FFN utilizes software CPU prefetching ($builtin\_prefetch$) to overlap loading from DRAM to the CPU cache with computations. On GPU, Spark FFN utilizes customized CUDA kernel.

The same masking vector also identifies the zero entries in the intermediate vector that is multiplied by matrix $V$ (Figure 4b). For this sparse vector-matrix multiplication, we store the matrix in row format. Each CPU thread (or GPU warp) processes a tile of the matrix while skipping the loading and computation of the masked rows. Prefetching and SIMD operations are applied similarly on CPU in this context.

Spark Attention utilizes these two types of sparse matrix multiplication operators to accelerate qkv computations for each head.

When extending to decoding with batch sizes greater than one or prefill, we continue to use individual masks to skip *computations* while using a union of masks from each vector within the batch to create unified masks for *memory loading*. With larger batches, Spark transformer is expected to save less memory loading (vs. original Gemma), unless there is significant overlap in top-k positions within the same batch. Nonetheless, the Spark transformer consistently reduces FLOP by skipping computations based on individual masks within the batch.

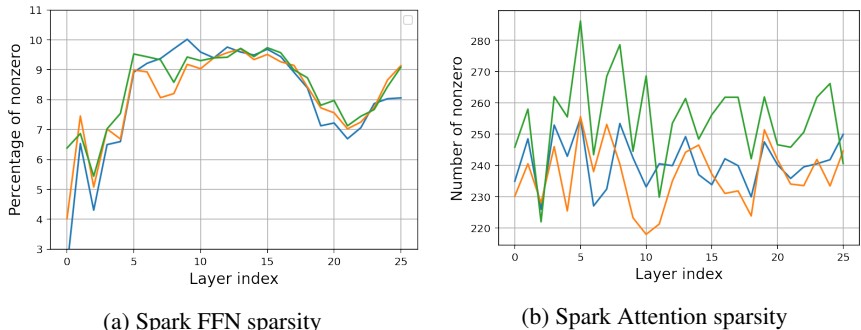

(a) Spark FFN sparsity          (b) Spark Attention sparsity

Figure C.1: Sparsity in the intermediate activation of Spark FFN and Spark Attention *during evaluation* (see Figure 3 for results during training). For FFN, we use a simple prompt "test" and report the percentage of nonzero entries in generating the 5th, 10th, and 15th token. For Attention, we report the nuber of nonzero entries at the 512th, 1024th, and 2048th token during prefill.

## Appendix C    Additional Experimental Results and Details

### C.1    Distribution of Inputs to Statistical Top-$k$

The underlying assumption for statistical top-$k$ is that the activation vector upon which it is applied to, namely, the pre-GELU activation in Spark FFN and the pre-softmax activation in Spark Attention, can be modeled as being drawn from an i.i.d. Gaussian distribution. Here we provide empirical evaluation on the distribution of these activation vectors for Spark Transformer. Results for Spark FFN and Spark Attention are provided in Figure C.4 and Figure C.5, respectively. The results show that the distribution holds close proximity to a Gaussian, hence justifying the use of statistical top-$k$.

### C.2    Sparsity Level During Evaluation

Complementing Figure 3 which reports sparsity level during pretraining, here we report the sparsity level during evaluation to confirm that statistical top-$k$ produces the same level of sparsity during test time. The results are presented in Figure C.1 for some arbitrarily selected tokens. For Attention, in particular, we select tokens at the positions 512, 1024, and 2048 which are all above our choice of $k = 256$ for Spark Attention.

### C.3    Batching Analysis

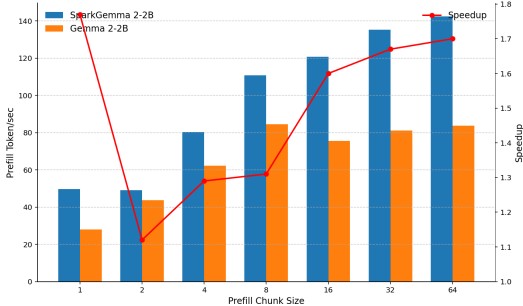

Figure C.2: Spark Transformer vs. Gemma-2 Prefill Token/Sec with Varying Chunk Size. We use a prompt length of 4096 tokens on a 16 core CPU VM.

Figure C.2 provides the performance comparison between Spark Gemma 2 and Gemma 2, measured in prefill throughput (tokens/sec) across varying chunk sizes. We use a 4096-token prompt on a 16-core CPU VM. A similar trend is expected during the decoding phase with varying batch sizes.

Our analysis shows that Spark Transformer provides the highest speedup at batch size 1, and again at large batch sizes (e.g. >8), where the compute FLOP becomes the primary bottleneck.

For Gemma-2, as seen in the figure, increasing batch/chunk size leads to a significant improvement in prefill throughput until the batch size reaches 8. This improvement occurs because batching reduces memory access by reusing weights across multiple tokens in the CPU cache. Once the computation becomes the bottleneck (i.e. batch = 8), further batching provides diminishing returns.

In contrast, Spark Transformer behaves differently. When the batch size increases from 1 to 2, we observe minimal throughput change. This is due to the lack of overlap in top-k positions between the tokens, resulting from the high sparsity. However, as the batch size increases beyond 4, Spark Transformer starts benefiting from weight reuse, similar to Gemma-2. Spark Transformer continues to show improvements in throughput until the batch size reaches approximately 64, where it eventually becomes FLOP-bound, much later than Gemma-2 due to the reduced FLOP requirements.

Overall, Spark Gemma demonstrates the most significant gains in two scenarios: when the batch size is 1, a common setting for desktop or mobile devices decoding, and when the batch size is large enough that FLOP becomes the dominant bottleneck.

## C.4 Effect of $r$ and $k$ in Spark FFN.

Spark FFN comes with two hyper-parameters, namely $r$ which controls the rank hence FLOP count of the low-cost predictor, and $k$ which controls sparsity of activation hence the FLOP count. In Figure C.3 we provide an ablation study on the effect of these two hyper-parameters, by reporting the training loss curves in the first 25,000 training steps (which is around 5% of full training). From Figure C.3a, the best choice of $r$ is 1024 which is nearly half of $d_{\text{model}} = 2304$ (due to model sharding constraint, $r$ cannot be taken to be exactly a half of $d_{\text{model}}$). From Figure C.3b, we see that the model quality is insensitive to choices of $k$ that gives $[5\%, 10\%]$ sparsity, but there is quality loss if we go sparser, e.g. 3% nonzeros.

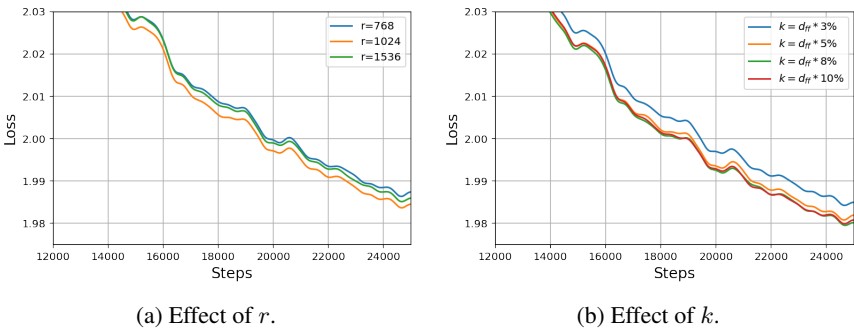

(a) Effect of $r$.              (b) Effect of $k$.

Figure C.3: Effect of hyper-parameters $r$ and $k$ in Spark FFN on training loss. A Gaussian filter of $\sigma = 200$ is applied to smooth the loss curves. Models are trained with 1/20 of standard Gemma-2 training iterations.

## C.5 Additional Ablation Studies

In this section, we provide ablation studies for understanding the effect of the individual components of Spark Transformer. Towards that, we plot the training loss curves for Gemma-2 and Spark Transformer, see Figure C.6. Here, we restrict to the first 80k training steps out of the 500k total steps since it is costly to fully train all ablation models, and that 80k steps is sufficient for seeing the trend. We can see that Spark Transformer slightly lags behind Gemma-2. However, as demonstrated in Figure 1b, that small difference in training loss does not lead to a substantial difference in evaluation quality.

In our ablation studies below, we add a single component at a time to Gemma-2 and report the quality impact.

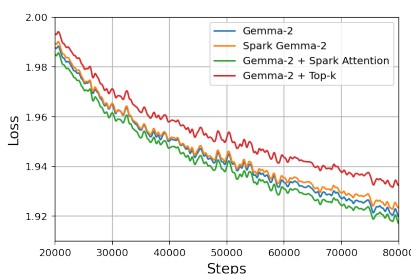

Figure C.6: Ablation study in terms of training loss in the first 80k training steps (out of 500k total steps).

**Spark FFN vs Spark Attention.** To understand the effect of Spark FFN vs Spark Attention, we conduct an experiment where only attention is switched from a standard one to Spark Attention,

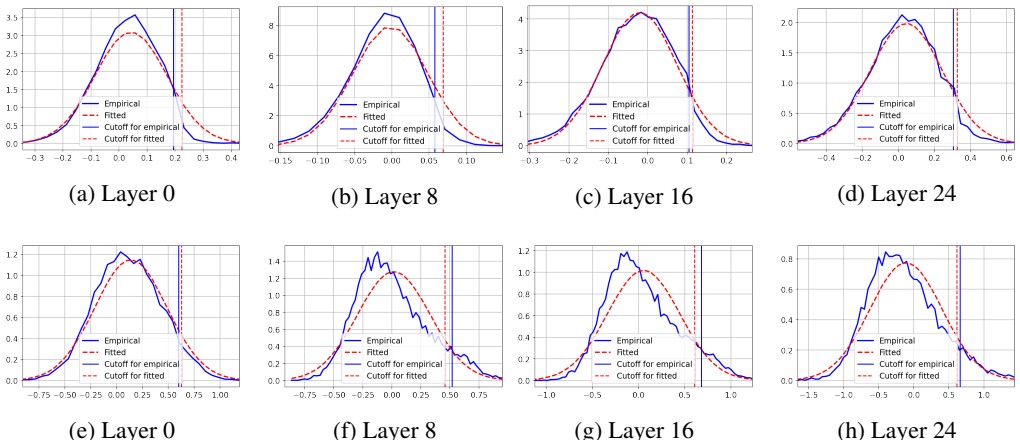

| (a) Layer 0 | (b) Layer 8 | (c) Layer 16 | (d) Layer 24 |

| (e) Layer 0 | (f) Layer 8 | (g) Layer 16 | (h) Layer 24 |

Figure C.4: Distribution of the entries of the input activation to statistical top-$k$ in Spark FFN (see Figure C.5 for result of Spark Attention). The two rows correspond to activation at two positions 0 and 1000 of an input, and the columns correspond to activation at four different depth levels $\{0, 8, 16, 24\}$ of the 26-layer pretrained Spark Transformer. The input is the first 1000 tokens of the first essay from `https://paulgraham.com/articles.html` prepended with the BOS token. We compare the empirical distribution (*Empirical*) with the Gaussian distribution whose mean and standard deviation (std) are computed as the sample mean and std of the input (*Fitted*). *We see that the Gaussian closely approximates the empirical distribution.* We also compare the cutoff value estimated from the Gaussian, i.e., $\theta(\boldsymbol{x}, k)$ used in eq. (10) with $k/d = 5\%$ (*Cutoff for fitted*), with the cutoff value for obtaining $8\%$ nonzeros on the empirical distribution (*Cutoff for empirical*). *It can be seen that these two cutoff values are close.*

whereas the FFN remains the standard one. The result is illustrated as *Gemma-2 + Spark Attention* in Figure C.6. It can be seen that Spark Attention provides a minor quality gain over Gemma-2. In comparing *Gemma-2 + Spark Attention* with Spark Transformer, this also shows that further adding Spark FFN slightly hurts model quality. As noted above, such a small difference does not lead to substantial quality impact on the evaluation tasks. Hence, we conclude here that none of Spark FFN and Spark Attention has significant quality impact.

**Sparsity enforcing vs Low-cost predictor.** Sparsity enforcing via statistical top-$k$ and low-cost activation predictor are two relatively independent components of Spark Transformer. This means that, upon the standard Gated FFN (see eq. (5)) that is used in Gemma-2, which we rewrite here for convenience:

$$\text{Gated-FFN}(\boldsymbol{q}; \boldsymbol{K}_1, \boldsymbol{K}_2, \boldsymbol{V}) = \boldsymbol{V} \cdot \left( \sigma \left( \boldsymbol{K}_1^\top \boldsymbol{q} \right) \odot \left( \boldsymbol{K}_2^\top \boldsymbol{q} \right) \right), \qquad (\text{C.1})$$

we may choose to only apply statistical top-$k$ for enforcing sparsity, i.e.,

$$\text{Topk-Gated-FFN}(\boldsymbol{q}; \boldsymbol{K}_1, \boldsymbol{K}_2, \boldsymbol{V}) = \boldsymbol{V} \cdot \left( \sigma \left( \text{Statistical-Top}_k(\boldsymbol{K}_1^\top \boldsymbol{q}) \right) \odot \left( \boldsymbol{K}_2^\top \boldsymbol{q} \right) \right). \qquad (\text{C.2})$$

Note that applying a sparsifying function on the input to the nonlinear function $\sigma()$ as in eq. (C.2) is a common choice in the literature of sparse activations, e.g., [64, 81, 49]; the main difference between these works lies in the specific sparsity enforcing technique, see Table D.1 for a summary. In addition to the sparsifying function, Spark FFN also has another architectural change for the purpose of introducing a low-cost predictor. Here, we rewrite Spark FFN for ease of comparison with eq. (C.2):

$$\text{Spark-FFN}(\boldsymbol{q}; \boldsymbol{K}, \boldsymbol{V}, k, r) \stackrel{\text{def}}{=} \boldsymbol{V} \cdot \left( \sigma \left( \text{Statistical-Top}_k(\boldsymbol{K}_1^\top \boldsymbol{q}[:r]) \right) \odot \left( \boldsymbol{K}_2^\top \boldsymbol{q}[r:] \right) \right). \qquad (\text{C.3})$$

Analogous to FFN, we may also only add statistical top-$k$ to attention without the low-cost predictor, i.e., by switching from standard Attention in eq. (6) to the following:

$$\text{Topk-Attention}(\boldsymbol{q}; \boldsymbol{K}, \boldsymbol{V}) \stackrel{\text{def}}{=} \boldsymbol{V} \cdot \text{softmax} \left( \text{Statistical-Top}_k^{(-\infty)}(\boldsymbol{K}^\top \boldsymbol{q}) \right). \qquad (\text{C.4})$$

Here, we aim to understand the effect of introducing statistical top-$k$ without the low-cost predictor. Towards that, we conduct an experiment where FFN and Attention in Gemma-2 are replaced with eq. (C.2) and eq. (C.4), respectively. The result is illustrated as *Gemma-2 + Top-k* in Figure C.6.

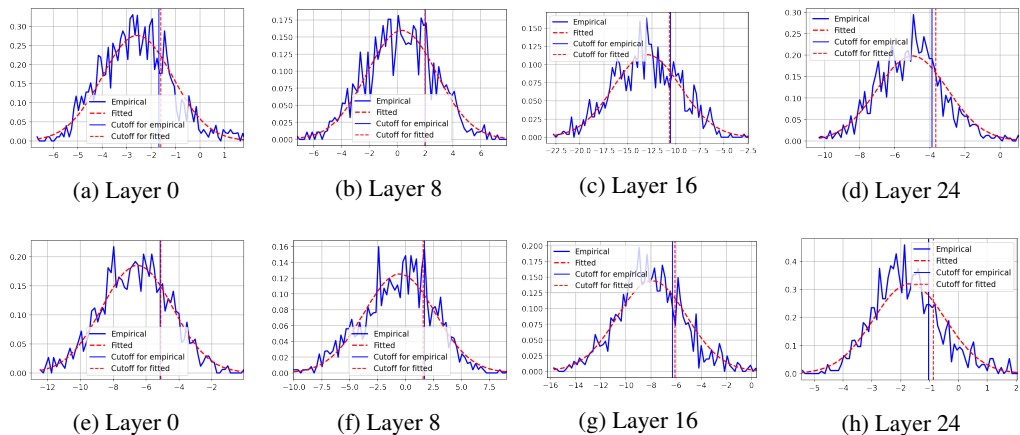

|     |     |     |     |
| :-: | :-: | :-: | :-: |
| (a) Layer 0 | (b) Layer 8 | (c) Layer 16 | (d) Layer 24 |
| (e) Layer 0 | (f) Layer 8 | (g) Layer 16 | (h) Layer 24 |

Figure C.5: Distribution of the entries of the input activation to statistical top-$k$ in Spark Attention (see Figure C.4 for result of Spark FFN). The two rows correspond to activation for two different attention heads, and the columns correspond to activation at four different depth levels $\{0, 8, 16, 24\}$ of the 26-layer pretrained Spark Transformer. Model input is the first 1000 tokens of the first essay from `https://paulgraham.com/articles.html` prepended with the BOS token, and we examine activation of the last token (i.e., inner product between the query embedding of the 1001st token and all 1001 key embeddings). We compare the empirical distribution (*Empirical*) with the Gaussian distribution whose mean and standard deviation (std) are computed as the sample mean and std of the input (*Fitted*). *We see that the Gaussian closely approximates the empirical distribution.* We also compare the cutoff value estimated from the Gaussian, i.e., $\theta(\boldsymbol{x}, k)$ used in eq. (10) with $k = 256$ (*Cutoff for fitted*), with the cutoff value for obtaining top 256 entries on the empirical distribution (*Cutoff for empirical*). *It can be seen that these two cutoff values are close.*

Table C.1: **FLOPs per token comparison: Spark Transformer vs. standard Transformer.** In a standard Transformer with model dimension $d_{\mathrm{model}}$, we assume multi-head attention where the sum of head dimensions equals $d_{\mathrm{model}}$, and an FFN with non-gated activation and width $d_{\mathrm{ff}}$. Here, $n_{\mathrm{ctx}}$ represents the context length for the target token. The computational cost is primarily determined by the FFN (assuming $d_{\mathrm{ff}} \gg d_{\mathrm{model}}$, which is typical) and the attention dot product (assuming a long context length). Spark Transformers introduce sparsity parameters, $k_{\mathrm{ff}}$ and $k_{\mathrm{attn}}$, to reduce FLOPs. Setting $k_{\mathrm{ff}} = 8\% \times d_{\mathrm{ff}}$ and $k_{\mathrm{attn}} = 256$ achieves a 3.2× FLOPs reduction in the FFN, a 4× reduction in the attention dot product, and a 2.5× reduction overall (assuming $n_{\mathrm{ctx}} = 8k$) for Gemma-2B.

| Operation | FLOPs per Token[7] | |
| :--- | :---: | :---: |
| | Standard Transformer | Spark Transformer (Ours) |
| FFN | $4d_{\mathrm{model}}d_{\mathrm{ff}}$ | $d_{\mathrm{model}}d_{\mathrm{ff}} + 3d_{\mathrm{model}}k_{\mathrm{ff}}$ |
| Attention dot product | $4d_{\mathrm{model}}n_{\mathrm{ctx}}$ | $d_{\mathrm{model}}n_{\mathrm{ctx}} + 3d_{\mathrm{model}}\min\{k_{\mathrm{attn}}, n_{\mathrm{ctx}}\}$ |
| Attention linear projection | $8d_{\mathrm{model}}^2$ | $8d_{\mathrm{model}}^2$ |

It can be seen that the training loss becomes notably larger and the gap compared to Gemma-2 is further increasing with more training steps. This result demonstrates that while the low-cost predictor is introduced to Spark Transformer for reducing the cost in predicting the nonzero entries, it also helps in bridging the gap from the introduction of statistical top-$k$. In other words, Transformer with low-rank predictors in FFN and Attention is more amenable to activation sparsification without quality loss.

## C.6 Computing FLOPs per Token

To understand the FLOPs saving reported in Figure 1a, we provide a comparison between FLOPs of the major components of a Transformer, including FFN, Attention dot product, and Attention projections, with that of Spark Transformer. The results are presented in Table C.1.

# Appendix D   Additional Discussions

## D.1   Additional Related Work

In the following, we review a few lines of work closely related to ours.

**Mixture of Experts (MoEs)** may be considered as a particular case of sparsely activated models which group the neurons in FFN and activate all neutrons in selected groups [78, 52]. Neuron grouping has the benefit of being better suited for training accelerators compared to unstructured activation sparsity. However, training of MoEs incurs extra complexities in algorithmic design and requires special hardware support [29]. Moreover, the structured nature of sparsity limits the model's flexibility and expressiveness, and recent work advocates the use of a larger number of smaller experts [18, 36]. On the other hand, the discovery of the naturally emerging unstructured activation sparsity has motivated the perspective of naturally emerging experts [91, 25, 16, 70, 85, 95].

**Sparse activation** is common approach to improve the efficiency of large models and many techniques for a low-cost activation prediction have been developed over the years, such as low-rank factorization [20], quantization [11], product keys [46], hashing [12], etc. With the popularity of modern Transformer models, these techniques become natural choices [40, 90, 58, 82] for reducing their high computation costs. In particular, a lot of the excitement comes from the discovery that the activations in FFNs are naturally sparse [91, 54, 61] and hence efficiency with activation sparsity is obtained without a quality toll.

Our work falls into the category of the latest work in this direction that aims to bring the benefits to the latest generation large language models that do not have natural sparsity. Early attempts [64, 69, 92] seek to bring back sparsity by switching back to ReLU variants, but it usually incurs a quality loss. The quality gap may be largely bridged by more careful tuning, but the activation becomes less sparse (e.g. $25\%$ nonzeros in LLAMA 7B [81]). Top-k has become a more popular choice for obtaining sparsity recently [83] and is able to maintain neutral quality while offering strong sparsity, but only in selected layers [89]. Moreover, such methods require finetuning to bring sparsity and also obtain a predictor. Without doing finetuning, [49, 56, 94] obtained at most $50\%$ nonzeros under neutral quality. In contrast to these works, our work not only obtains $8\%$ nonzeros in activation of all FFN layers, but also a predictor, all with a single-stage training. We provide a summary of comparison to these methods in Section D.2.

Finally, the usefulness of activation sparsity goes beyond efficiency. For example, theoretical studies show its benefits for model generalizability and learnability [65, 6]. Moreover, activated neurons may be associated with semantic concepts, which offers understanding of the working mechanism and enables manipulating the output of Transformer models [17, 60].

**Sparse attention** broadly refers to the approach of attending to a selected subset of tokens in the context as a means of reducing computation cost [21, 42, 73]. Works on sparse attention include those that use handcrafted attention patterns [15, 9, 2, 24], which feature simplicity, and learned attention patterns [45, 75, 66] which feature better modeling capacity. However, learning attention patterns often involve learning, e.g., a hash table or k-means centers, which significantly complicates modeling. Closely related to our Spark Attention is the top-$k$ attention [35], which obtains data-adaptive attention simply from top-$k$ thresholding. Our work improves upon top-$k$ attention by introducing a low cost predictor which enables an increased computational benefits from sparsity. Finally, KV pruning approaches drop selected tokens permanently as decoding proceeds [93, 57], and cannot achieve as high compression ratio as sparse attention based approaches.

## D.2   Comparison with Related Work on Activation Sparsity in FFN

In Table D.1, we provide a summary of recent work on enabling FFN activation sparsity in the latest LLMs, including ReLUification [64], ProSpare [81], HiRE [89], and CATS [49]. Please see Section 5 for a discussion of these methods.

We can see that our Spark Transformer leads to a FLOPs reduction of -72%, which is more than all the other methods. This comes at the cost of only a -0.9% quality loss, which is lower than most of the other methods except HiRE. In particular, HiRE is on par with ours in terms of quality loss but achieves less FLOPs reduction.

Table D.1: Comparison with related work on enforcing activation sparsity in FFN of LLMs. Spark Transformer has the largest FLOPs reduction with one of the smallest quality loss.

| | Main Techniques | | Main Results | | | | |
|---|---|---|---|---|---|---|---|
| | Enforce sparsity | Predict support | Base model | Training cost[8] | Sparsity (%zeros) | Quality[9] | FFN FLOPs |
| ReLUification[10] [64] | ReLU | None | OPT 1.3B | +0% | 93% | -2% | -62% |
| | ReLU | None | Falcon 7B | +2% | 94% | -2.5% | -62% |
| | | | Llama 7B | +3% | 62% | -1.9% | -42% |
| ProSparse [81] | ReLU + $\|\cdot\|_1$ | None | Llama2 7B | +1.8% | 88% | -1.1% | -59% |
| | | | Llama2 13B | +6.7% | 88% | -1.4% | -59% |
| | | 2-layer FFN | Llama2 7B | NA | 75% | NA | NA |
| | | | Llama2 13B | NA | 78% | NA | NA |
| HiRE [89] | Group top$_k$ + commonpath | Low-rank / quantization | PALM2 1B | +100% | 80% | -0.8% | -60%[11] |
| CATS [49] | Thresholding | None | Mistral 7B | +0% | 50% | -1.5% | -33% |
| | | | Llama2 7B | | 50% | -2.4% | -33% |
| Spark Transformer | Statistical-top$_k$ | Partial dimensions | Gemma 2B | +0% | 92% | -0.9% | -72% |

**Comparing pretraining, finetuning, and "zero-shot" approaches.** Existing approaches on enforcing activation sparsity in FFNs can be categorised into three groups depending on when the sparsity is enforced during the training process.

- **Pretraining-based** approaches are those whose activation sparsity is enforced from the very beginning of the model pretraining, usually with certain architectural changes upon an established model. Hence, such methods require pretraining a model from scratch. For example, ReLUification [64] is the method of switching the activation function in FFN from a commonly used one, e.g. GELU, back to ReLU. When tested on OPT 1.3B, [64] applied this modification and pretrained a version of OPT 1.3B with ReLU from scratch (Table D.1 contains a summary of relevant results for it).

- **Finetuning-based** approaches refer to those that take an existing pretrained model without activation sparsity, and perform additional training steps usually with certain architectural changes to induce sparsity. For example, ReLUification discussed above as a pretraining-based approach can also be used as a finetuning-based approach by switching the activation function of a pretrained model to ReLU and performing additional training steps. In [64], this approach is applied to Falcon 7B and Llama 7B, for which the results are summarized in Table D.1. Other methods falling into this category, which we have summarized in Table D.1, include ProSparse [81] and HiRE [89].

- **"Zero-shot"** approaches. This refers to methods that enforces activation sparsity upon a pretrained model, without any additional training. CATS [49] is a method of this kind.

Our Spark Transformer belongs to the category of pretraining-based method, that is, it requires pre-training a model from scratch. It may not be used directly as a finetuning-based approach, due to the drastic difference between the architecture of a standard FFN with gated activation, i.e. eq. (5), and our Spark-FFN. This appears to make Spark Transformer less favorable than finetuning-based approaches, as the latter requires usually a small fraction of the pretraining steps in the finetuning process, see Table D.1 for a summary. However, the long-term serving costs of such models usually dominate overall expenditures, in which case the additional pre-training costs are amortized by the significant efficiency benefits realized from a higher level of sparsity and FLOPs reduction during model inference.

### D.3 Comparison of Statistical Top-$k$ with Related Top-$k$ Operators

The variational form in eq. (15) reveals connections of $\mathrm{Statistical}\text{-}\mathrm{Top}_k$ with other top-$k$ algorithms in the literature. Specifically, [51] defines a *soft top-$k$* as

$$\underset{\boldsymbol{z}}{\arg\min} \quad -\theta \cdot H(\boldsymbol{z}) - \langle \boldsymbol{z}, \boldsymbol{x} \rangle, \ \ \text{s.t.} \ \ \boldsymbol{z}^\top \mathbf{1} = k, \ \ \mathbf{0} \leq \boldsymbol{z} \leq \mathbf{1}, \tag{D.1}$$

where $H(\boldsymbol{z})$ is the entropy function. Another work [59] defines the *SparseK* operator

$$\underset{\boldsymbol{z}}{\arg\min} \quad -H^G(\boldsymbol{z}) - \langle \boldsymbol{z}, \boldsymbol{x} \rangle, \ \ \text{s.t.} \ \ \boldsymbol{z}^\top \mathbf{1} = k, \ \ \mathbf{0} \leq \boldsymbol{z} \leq \mathbf{1}, \tag{D.2}$$

where $H^G(\boldsymbol{z})$ is the generalized Gini entropy.

$\mathrm{Statistical}\text{-}\mathrm{Top}_k$ in the form of eq. (15), as well as eq. (D.1) and eq. (D.2), can all be interpreted as finding an output that is *close* to the input subject to a sparsifying regularization. Their major difference lies in the choice of the sparse regularization. That is, soft top-$k$ and SparseK uses entropy and Gini entropy, respectively, whereas $\mathrm{Statistical}\text{-}\mathrm{Top}_k$ uses $\ell_1$. The choice of $\ell_1$ makes $\mathrm{Statistical}\text{-}\mathrm{Top}_k$ superior in that it has a closed form solution provided by soft-thresholding, which only requires $d$ FLOPs. In contrast, soft top-$k$ and SparseK both do not have closed form solutions and require an iterative algorithm with a FLOP count dependent on the number of iterations. In addition, there is no guarantee that soft top-$k$ and SparseK can obtain (approximately) $k$ nonzero entries as output.

Finally, we mention that ideas similar to statistical top-$k$ have been used [79, 62] for the problem of distributed training [55]. However, we are the first to introduce, adapt, and verify its effectiveness for activation sparsity. Additional discussions are provided in Section D.4.

### D.4 Additional Discussions on Statistical Top-$k$

**Novelty upon existing work.** We note that ideas similar to our statistical top-$k$ have appeared in the literature. In particular, [79] introduced the idea of fitting a Gaussian distribution to the entries of an input vector and estimating a threshold from quantile functions. Then, [62] extended the approach to additional distributions. In both cases, the method is used for solving the problem of distributed training. Here, we summarize our contribution upon these works:

- We are the first to use statistical top-$k$ for enforcing activation sparsity in Transformers. Improving Transformer efficiency via activation sparsity has become a very popular research topic (see Section 5), but may have been suffering from a lack of efficient top-$k$ algorithms for enforcing sparsity. Hence, the introduction of statistical top-$k$ may facilitate the development of this area.

- Synergizing statistical top-$k$ into Transformers is nontrivial. Since the method of statistical top-$k$ is based on fitting a statistical distribution to the activation vector, there is the need to understand the distributions of different activations in order to determine which particular activation vector is suited for the application of statistical top-$k$ and the associated choice of the distribution. In our case, we decide that statistical top-$k$ should be applied to the activation before the nonlinear function (for FFN) and before softmax (for Attention) since entries of this vector provably follow a

---

[7]Please refer to Section 2.1 and Section 2.2 for the calculation of FLOPs for FFN and Attention, respectively. We omit non-leading-order terms (e.g., those arising from embedding, normalization, and nonlinear layers) and exclude the number of layers as a common multiplier.

[8]Total training cost relative to the base model. For finetuning based approaches, such as ReLUification (on Falcon and Llama) and ProSparse, the total training cost includes both the pretraining cost and the finetuning cost.

[9]Quality loss relative to the base model. Here the numbers are based on the results reported in their respective papers. Note that a different set of evaluation benchmarks is used in each paper. For ReLUification, this set contains ARC-Easy, HellaSwag, Lambada (for OPT) and Arc-E, Arc-C, Hellaswag, BoolQ, PIQA, LAMBADA, TriviaQA, WinoGrande, SciQ (for Falcon and Llama). For ProSparse, this set contains HumanEval, MBPP, PIQA, SIQA, HellaSwag, WinoGrande, COPA, BoolQ, LAMBADA, and TyDiQA. For HiRE, this set contains WMT14/WMT22, SuperGLUE, Multiple QA datasets, and multiple discriminative tasks datasets. For CATS, this set contains OpenBookQA, ARC Easy, Winogrande, HellaSwag, ARC Challenge, PIQA, BoolQ, and SCI-Q. For Spark Transformer, the datasets are those reported in Figure 1b.

[10]Results reported here are for the stage 1 training of their paper.

[11]Specifically, -80% on odd layers only, and -60% on average.

Gaussian distribution at random initialization. We also verify empirically that statistical top-$k$ is still reliable even after initialization.

- We extended statistical top-$k$ from using the hard-thresholding operator with the estimated statistical threshold to the soft-thresholding operator. This leads to a continuous optimization landscape that may have facilitated the optimization. Empirically, we found this choice to provide quality benefits for Spark Transformer.

- We provide the first theoretical justification for the correctness of statistical top-$k$, see Theorem 3.1.

- We reveal the conceptual connection between statistical top-$k$ and several related top-$k$ operators in the literature, see Section D.3. Such connections may motivate the development of more powerful top-k algorithms in the future.

**Handling cases when the activation is sharded.** The training of modern large Transformer models usually requires sharding certain model weights and activations across multiple computation devices, due to physical limitations on each device's memory. In particular, if sharding is used for the activations upon which the statistical top-$k$ is applied to, i.e., the pre-GELU activation in FFN and the pre-softmax activation in attention, extra care needs to be taken so that statistical top-$k$ is applied correctly. While this has not been the case for our experiment on Gemma-2 2B, here we discuss possible solutions if this case arises, e.g., when training a larger Spark Transformer for which sharding relevant activations may become necessary.

Assume that an activation vector of length $N$ is sharded over $m$ devices, and we are interested in finding approximately the top-$k$ entries of $N$ with the largest value. There are two ways of applying statistical top-$k$ for this purpose.

- Global statistical top-$k$. Here we require each device to compute the mean and variance for entries on itself, then communicate them to all other devices. In this case, each device receives $m - 1$ mean and variance values, which can be combined with mean and variance on its own to obtain global mean and global variance. Then, the rest of the steps in statistical top-$k$ can be carried out on individual devices. In this method, the output is exactly the same as if applying statistical top-$k$ without activation sharding. In terms of cost, there is extra computation coming from aggregating mean and variance from all devices, but the cost is very low as it requires only $O(k)$ FLOPs. The method also introduces a communication cost, but the cost is again small as each device only needs to send / receive $2k - 2$ floating point numbers.

- Local statistical top-$k$. Here we simply apply statistical top-$k'$ to entries on its own device with $k' = k/m$. The method is sub-optimal in the sense that it does not necessarily produce the same output as applying the global statistical top-$k$. However, it has the benefit of not adding any computation and communication cost.

In cases where $k \ll N$, the global statistical top-$k$ above is cheap enough and hence could be a natural choice.

### D.5 Synergy with Speculative Decoding

Speculative decoding is a prominent technique for accelerating the inference of large autoregressive models [53, 13]. It employs a smaller, faster "draft model" to generate a block of candidate tokens, which are then verified in a single, parallel forward pass by the larger, more accurate "target model". We view Spark Transformer as a highly complementary and synergistic technology, as its architecture enhances efficiency in both roles within the speculative decoding framework.

**Spark Transformer as the target model.** The primary computational bottleneck in speculative decoding is the parallel verification step, where the target model processes multiple tokens simultaneously. Spark Transformer is ideally suited to accelerate this step.

As the target model, Spark Transformer directly reduces the latency of this bottleneck, leveraging the same wall-time speedups demonstrated in single-token decoding. While verifying multiple tokens in parallel (e.g., $k = 2$ to $k = 4$) necessarily activates more neurons than a single-token pass, significant sparsity is still maintained. This is for two primary reasons:

- Small Draft Size: The number of speculative tokens is typically small, meaning the union of activated neurons remains a small fraction of the total.

- Aggregated Sparsity: Consecutive tokens in a natural language sequence often share a substantial portion of their activated neurons. This phenomenon, termed aggregated sparsity [64], is an extension of the lazy neuron phenomenon [54] and the observed persistence of activation patterns for related inputs.

Because the union of activated FFN neurons and attended tokens remains sparse, Spark Transformer effectively reduces the computational cost of the most expensive part of the speculative decoding loop.

**Spark Transformer as the Draft Model.** An effective draft model must satisfy two criteria: it must be significantly faster than the target model, and it must be accurate (i.e., its predictions must frequently align with the target model's). High accuracy is crucial as it dictates the token acceptance rate, which is the primary driver of the overall speedup.

Spark Transformer is a strong candidate for a high-performance draft model. Our results demonstrate that it achieves near-quality neutrality with its dense counterpart while operating at a fraction of the inference cost. Using a Spark Transformer as the draft model — either a smaller version or even the same model with a lower sparsity budget — could lead to a much higher acceptance rate than a standard distilled model of equivalent speed. This high fidelity, combined with low latency, makes it an ideal driver for maximizing the efficiency of the speculative decoding process.

### D.6 Synergy with Quantization Techniques

We view Spark Transformer and quantization as two orthogonal and highly synergistic optimization axes. Quantization, such as conversion to FP8 or INT8, reduces the computational cost and memory size of each individual operation and parameter. Spark Transformer, in contrast, reduces the total number of operations and memory accesses by dynamically pruning FFN activations and attention tokens. Critically, our sparse implementation also reduces the memory bandwidth bottleneck by skipping the load of masked weights from HBM. The combined benefits are therefore expected to be multiplicative, leading to significant compounded gains in inference throughput and energy efficiency.

A key question is whether a model that is already sparsely activated is more sensitive to the precision loss from quantization. We hypothesize that, for activation quantization in particular, Spark Transformer may reduce the model's sensitivity and be less prone to accuracy degradation compared to a dense model. This is critical, as recent work has shown that standard sparsity and quantization can be in contradiction: magnitude-based sparsity preserves large-value outliers that are detrimental to quantization-aware scaling [34]. This hypothesis stems directly from the design of our statistical top-$k$ operator.

- Dynamic Range Compression: The statistical top-$k$ operator in Spark Transformer employs soft-thresholding. This function outputs $\max(x - \theta(x, k), 0)$, effectively subtracting the learned threshold $\theta(x, k)$ from all activated neurons. This operation inherently compresses the dynamic range of the activation tensor. A primary challenge in LLM activation quantization is the presence of large-magnitude "outlier" values, which skew the quantization scaling factor and lead to significant precision loss for the more common, smaller values [22]. By "pre-conditioning" the distribution and shrinking these outliers, our operator makes the activation tensor significantly more amenable to accurate low-bit representation.
- Zero-Point Stability: The 92% of FFN neurons that are not in the top-k are set to 0. This large volume of true zeros is perfectly and losslessly represented in any quantization format, avoiding the "near-zero" noise that can plague dense models.

For weight quantization, we hypothesize that the sensitivity would be broadly similar to the dense Gemma-2 counterpart, as Spark Transformer reallocates and reuses the full set of parameters rather than pruning them.

While a full empirical study of this interaction is a highly promising direction for future work, we hypothesize that Spark Transformer is not only compatible with quantization but may actively facilitate it. This positions our work alongside other recent efforts to create novel sparse-quantized representations [23], offering a robust path to further acceleration."

