# OpenReview forum: "Spark Transformer: Reactivating Sparsity in Transformer FFN and Attention"
_NeurIPS.cc/2025/Conference — NeurIPS 2025 poster_

### Official Review · Reviewer_1Jfq · 2025-06-22

**Clarity:** 4
**Significance:** 3
**Originality:** 3
**Rating:** 5
**Confidence:** 3

**Summary:**

The proposed Spark Transformer consists of two parts: Spark FFN and Spark Attention. The authors dedicate/repurpose a portion of hidden states to estimate the importance of weights and KV vectors against the given hidden state (e.g., K and V vectors). Further, Spark Transformer identifies top-K elements in an approximate manner, not requiring the costly process of sorting. On the CPU and low-profile GPU machines, the proposed Spark Transformer shows notable speedup ranging from 1.25x to 1.79x.

**Questions:**

* Can you further articulate the interaction of Spark Transformer and speculative decoding?
* Can you further quantify or qualitatively discuss the effectiveness of Spark Transformer on conventional serving environments on the cloud?
* Can you discuss the interplay between Spark Transformer and quantization techniques? (e.g., sensitivity of its accuracy while using different data types.)

Not questions, but minor comments:
*  Section vs. section: be consistent.
* “an” NVIDIA GPU

**Ethical Concerns:**

["NO or VERY MINOR ethics concerns only"]

**Final Justification:**

Interaction with the authors and the reviews of the fellow reviewers help me keep my initial positive score unchanged.

**Limitations:**

yes

**Paper Formatting Concerns:**

None.

**Quality:**

4

**Strengths And Weaknesses:**

Strengths:
+ The core ideas of embedding sparsity information within hidden states and avoiding the cost of sorting by approximately estimating top-k elements are well taken.
+ The paper is well written and structured organizationally.
+ Presenting results for CPUs and GPUs is another plus.

Weaknesses:
- Spark Transformer seems to target low-profile devices, where the degree of batching is not likely high (e.g., mobile devices). In this case, speculative decoding is gaining popularity, where the effective batch size of a ‘target’ model  increases. This paper does not even mention speculative decoding though.
- It is not clear if the proposed idea can be highly effective to the cloud environments, which use more powerful GPUs/TPUs and where the batch size is neither extremely high (violating goodput in servers) or low (running without speculative decoding in mobile devices), thus, it would be limited by the memory bandwidth of fetching weights from main memory (e.g., HBM). In these scenarios, saving computation might not lead to any considerable improvement in overall performance and energy efficiency as the serving system is memory-bound and communication dominates computation in energy consumption. More discussion in these deployment scenarios will further strengthen the paper.

---

> ### Author Rebuttal · Authors · 2025-07-31
>
> We thank Reviewer 1Jfq for the excellent review and the insightful, forward-looking questions about our work's practical deployment. We are encouraged that the reviewer recognized the novelty of our core ideas and the quality of our presentation.
>
> # **Interaction with Speculative Decoding**
>
> Thank you for raising this highly relevant point. We view Spark Transformer and speculative decoding as **highly complementary and synergistic technologies**. Our method enhances both roles within the speculative decoding framework.
>
> - As the Target Model: Spark Transformer directly accelerates the costly verification step. While verifying multiple tokens at once means more neurons are activated compared to single-token decoding, significant sparsity is still maintained for two reasons: 1) The number of speculative tokens is typically small (e.g., 2-4), and 2) Consecutive tokens often share activated neurons (a phenomenon termed "Aggregated Sparsity"). This allows Spark Transformer to **reduce the latency of the primary bottleneck** in the speculative decoding loop.
> - As the Draft Model: A great draft model is both fast and accurate. Because Spark Transformer nearly matches the quality of its dense counterpart at a fraction of the inference cost, it serves as an **ideal high-performance draft model**, potentially leading to higher token acceptance rates and greater overall speedups.
>
> We will add a detailed discussion of this synergy to the revised manuscript.
>
> # **Effectiveness in Memory-Bound Cloud Environments**
>
> This is a critical point for practical deployment. While LLM inference can indeed be memory-bound, our approach is specifically designed to provide substantial benefits in this regime by **reducing both computation and memory traffic**.
>
> Our innovation is not just a theoretical FLOP reduction. As detailed in Appendix B and illustrated in Figure 4, our custom sparse kernels are implemented to **avoid loading the weights corresponding to zeroed-out activations from High-Bandwidth Memory (HBM)**. Therefore, our activation sparsity translates directly into **reduced memory I/O**.
>
> This means our reported ~2.5x FLOP reduction is coupled with a significant reduction in memory access, directly addressing the memory bandwidth bottleneck. This dual benefit also leads to **substantial energy savings**, a vital metric for large-scale cloud deployments. We will clarify this crucial point more explicitly in the revised paper.
>
> # **Interplay with Quantization**
>
> This is another excellent suggestion. We consider activation sparsity and weight quantization to be **orthogonal optimization techniques** that can be combined for even greater efficiency.
>
> While we have not yet performed these experiments, we see no fundamental conflicts. Standard methods like post-training quantization (PTQ) or quantization-aware training (QAT) should be directly applicable to Spark Transformer. We hypothesize that because our model learns activation importance scores during training, it may even prove robust to the precision loss from quantization. Investigating this powerful combination is a promising avenue for future work, and we will add it to our discussion section.

---

### Official Review · Reviewer_PzjZ · 2025-06-25

**Clarity:** 2
**Significance:** 3
**Originality:** 3
**Rating:** 4
**Confidence:** 3

**Summary:**

1  Summary
The paper proposes Spark Transformer, an architecture that re‑introduces and exploits activation sparsity in both feed‑forward networks (FFN) and multi‑head attention. The key technical pieces are
* Spark FFN – splits the input,  locate the k most relevant neurons via the "statistical top‑k operator", and performs the remaining matmuls sparsely.
* Spark Attention – mirrors the same idea for query–key scores, limiting each token to at most 256 attended tokens.
* Statistical top‑k – a linear‑time, differentiable approximation of top‑k based on Gaussian quantiles coupled with soft‑thresholding; analysis (Thm 3.1/3.2) bounds the expected error and shows differentiability.
* An implementation on Gemma‑2 (2 B) shows ≈ 8 % active FFN neurons and ≤ 256 attention edges, yielding 2.5× theoretical FLOP reduction and 1.86× CPU / 1.25–1.40× GPU decode speed‑ups with negligible quality loss on standard LLM benchmarks.

**Questions:**

1. GPU speed‑up baseline – Could you provide absolute tokens/s for Spark versus a dense Gemma‑2 implementation on the same T4, so the community can reproduce the 1.25–1.40× figure? (Fig. 5’s GPU bars compare Spark without optimization  vs. Spark with optimization, not Spark vs. dense Gemma‑2)
2. Huber versus ReLU – Have you tried dropping Huber (δ → 0) or replacing it with the ReLU‑style hard threshold? How does this affect convergence and final quality?
3. Predictor overhead comparison – When integrating a Deja‑Vu‑style low‑rank predictor into a standard FFN (no splitting), what is the FLOP/latency trade‑off relative to Spark FFN?
4. Attention sparsity generality – Did you evaluate Spark Attention on tasks requiring long‑range reasoning (e.g., PG‑19, LongBench) to ensure the 256‑edge cap does not hurt quality?
5. Training stability – Were any gradient‑scaling or learning‑rate adjustments needed because of statistical top‑k, or does the recipe match Gemma‑2 exactly?

**Ethical Concerns:**

["NO or VERY MINOR ethics concerns only"]

**Final Justification:**

Thank you for the response and additional results.
Now all of my concerns have been resolved.
The idea of utilizing activation sparsity w/o prediction is very interesting and practical.
I keep my original rating.

**Limitations:**

yes

**Quality:**

3

**Strengths And Weaknesses:**

## Strengths
1. Novel formulation utilizing activation sparsity which does not requires  Deja-vu-style "predictor".
2. Linear‑time top‑k backed by theory and ablation showing <1 % training slow‑down versus >10× for JAX approx‑max‑k.
3. End‑to‑end pre‑training at scale (2 T tokens) with near‑neutral accuracy on 8 diverse downstream tasks.
4. Evidence of wall‑time wins on commodity 16‑core CPUs and a low‑profile NVIDIA T4 GPU.

## Weaknesses
1. GPU baseline ambiguity – Fig. 5 compares “optimized vs. un‑optimized” Spark kernels; the paper does not show a direct comparison against a dense Gemma‑2 baseline, making the reported 1.25–1.40× speed‑up hard to interpret.
2. Regularisation ablation missing – Sec. 3.2 advocates Huber regularisation for smoothing statistical top‑k, but no experiment contrasts Huber with the common ReLU (or no) regulariser to isolate its benefit.
3. Comparison to predictor‑based sparse methods – the paper qualitatively cites Deja‑Vu, PowerInfer, etc., yet does not provide a quantitative comparison on speed‑up vs. additional parameters / training cost, making it hard to gauge relative efficiency.

---

> ### Author Rebuttal · Authors · 2025-07-31
>
> We thank Reviewer PzjZ for their thorough review and for recognizing the novelty of our formulation, the strength of our theoretical backing, and the evidence of wall-time wins. We address the specific technical questions below.
>
> # **Baseline Ambiguity in Figure 5**
>
> This is an excellent point regarding clarity. To provide a direct comparison, we have conducted new decoding speed evaluations. We benchmarked our optimized Spark Gemma-2 against the original dense Gemma-2 (on CPU, with ongoing GPU experiments), and also measured the overhead of our architecture before sparsity optimizations were applied.
>
> The results confirm a significant net speedup. The slight overhead from our operators is far outweighed by the gains from sparsity.
>
> | Platform           |  Dense Gemma-2 (Baseline) | Spark (No optimization) | Spark (Full optimization) |
> | -------------          | ---------------------------------------  |  -----------------------------    | --------------------------------- |
> | CPU (4-core)   | 1.00x                                         | 0.87x - 0.96x                  |  **1.42x - 1.64x**            |
> | CPU (16-core) | 1.00x                                       | 0.92x - 0.97x                  |  **1.35x - 1.79x**            |
> *Table. Decoding speed (tokens / sec) relative to the baseline*
>
> # **Huber Regularization Ablation**
>
> Thank you for this question. As briefly mentioned (L209), we empirically investigated this. We found that a non-zero hyperparameter δ for the Huber loss provided no noticeable improvement in model quality or training stability over setting δ=0. Therefore, for simplicity and to avoid introducing an unnecessary hyperparameter, we proceeded with δ=0 in all experiments, which effectively reduces to the standard soft-thresholding operator.
>
> # **Comparison to Predictor-Based Methods like Deja-Vu**
>
> This is a crucial point that distinguishes our work. Our approach targets a fundamentally different challenge than methods like Deja-Vu.
>
> - **Deja-Vu** was designed for older architectures (e.g., OPT) with naturally sparse activations. Its predictor identifies this pre-existing sparsity.
> - **Our work** targets modern LLMs with non-sparse gated activations (e.g., SwiGLU). Here, no natural sparsity exists. Our statistical operator is part of an architecture that induces structured sparsity from scratch while minimizing quality loss.
>
> # **Attention Sparsity on Long-Range Reasoning Tasks**
>
> This is an important consideration. Our work focuses on architectural innovation within the context of the base model's capabilities. Since the Gemma-2 models we build upon are pre-trained for an 8k context length, they do not generalize well beyond that window. Benchmarking on long-context tasks would therefore not yield meaningful results for either the dense baseline or our Spark Transformer.
>
> We agree that evaluating a version of Spark Transformer pre-trained for a long-context regime is a vital piece of future work, but it is beyond the scope of the current study.
>
> # **Training Stability and Recipe Adjustments**
>
> One of the practical advantages of our method is its stability. The Spark components integrated seamlessly with the standard Gemma-2 training recipe, **requiring no adjustments** to the learning rate, optimizer, or gradient scaling. This highlights the robustness and ease of adoption of our approach.

---

### Official Review · Reviewer_4xrP · 2025-06-29

**Clarity:** 3
**Significance:** 2
**Originality:** 2
**Rating:** 3
**Confidence:** 4

**Summary:**

The paper modifies the transformer architecture into a new Spark Transformer model, leveraging sparsity in the model activations through predictors that decide which modules in the model are activated during inference. Sparsity is introduced in the training process, without any additional post-training modifications to the model. To efficiently identify sparse parts of the model, the authors introduce a novel statistical top-k operator. Spark Transformer version of Gemma-2B performs close to the original model using 40% of its FLOPS and provides 1.8 inference speedup on CPU and 1.4 inference speedup on GPU.

**Questions:**

Why is the idea only evaluated for a single, relatively small LLM? The authors justification for not providing the training code (“Training data and code for Gemma-2 are proprietary.”) indicates that the authors should have access to sufficient computational resources. If the authors cannot provide the code, the paper should at least contain strong experimental evidence that allows the reader to adequately compare the proposed method to the alternatives and judge its impact of wider set of models. However, all the experiments are contained to a single architecture (Gemma 2B) and the authors do not perform sufficient experiments to compare existing alternatives to their method even in this single setting.

**Ethical Concerns:**

["NO or VERY MINOR ethics concerns only"]

**Final Justification:**

After the discussion phase, I am confident that the paper’s technical contributions are strong and that it provides a clear practical and technical impact. I have chosen to keep my score unchanged to reflect certain evaluation concerns (as mentioned in more detail during the discussion), but I am now convinced that the positive aspects of the work outweigh the negatives and justify acceptance.

**Limitations:**

Yes

**Paper Formatting Concerns:**

No concerns

**Quality:**

2

**Strengths And Weaknesses:**

# Strengths

1. The paper introduces a method to directly integrate activation sparsity into the model via jointly trained sparsity predictors, which avoids the need to apply any post-training modifications and leads to an elegant solution.

2. The proposed top-k operator is a novel and efficient alternative to standard top-k selection.

3. Unlike many prior works that focus solely on FFN sparsity, this paper applies activation sparsity to attention mechanisms as well.

4. The method supports efficient batched inference, as shown in Appendix C.3, which is not always given in sparse activation papers.

5. Writing is very clear and easy to follow.

# Weaknesses

1. The empirical results are limited to a single model (Gemma-2B), and there are no results that would provide evidence that the method generalizes across architectures, model sizes, or tasks.

2. The paper mentions related works on activation sparsity and provides a comparison between their approach and existing alternatives in Table D.1. However, this table contains only main results scraped from the other papers, not experimental results reproduced by the authors. Moreover, these results are not really compatible, as they refer to different models. Therefore, the paper does not provide sufficient results to convincingly position the method with respect to the other existing alternatives. Additionally, while wall-clock speedups for Spark Transformer are mentioned in the paper, the table comparing Spark with other methods includes only FLOPS, which do not directly translate to the inference efficiency.

3. Paper emphasizes CPU speedups and does not provide results for common GPUs like A100, which makes it hard to judge the practical impact of the method for the most common deep learning accelerators.

4. The paper gives a fairly detailed description of the architectural modifications, including hyperparameters, training setup, and implementation details. However, the authors do not share the code, which greatly affects the reproducibility of their work.

# Summary

While the paper proposes an interesting and potentially impactful idea, the lack of experimental evaluation makes it impossible to judge the impact of the proposed method in practice and judge whether it provides a significant efficiency improvements compared to already existing alternatives.

---

> ### Author Rebuttal · Authors · 2025-07-31
>
> We thank Reviewer 4xrP for their feedback. We appreciate the acknowledgment of our work's strengths, including the elegant end-to-end training, the novelty of the statistical top-k operator, and the application of sparsity to both FFN and Attention. We address the noted weaknesses below.
>
> **Empirical Evaluation using Gemma-2B only**
>
> We thank the reviewer for this suggestion. The decision to focus our study on Gemma-2B was driven by the substantial computational cost of pre-training. This approach allowed us to conduct a thorough and controlled investigation on a strong, representative open-weight model, a common practice in LLM pre-training research where resources are a key constraint.
>
> While larger models like Gemma-2 9B are an important future direction, their pre-training cost was beyond the scope of this project. By concentrating our resources, we were able to provide what is, to our knowledge, one of the first complete pre-training demonstrations of activation sparsity's feasibility in a modern LLM. We believe this foundational evidence is a valuable contribution and provides a strong empirical basis for future explorations on larger-scale models.
>
>
> **Code availability**
>
> While proprietary constraints on the full training data and scripts prevent a complete code release, we are committed to ensuring our work is reproducible. To this end, we have provided the following key components to the community:
> - A detailed architectural blueprint of our Spark Transformer (Section 2, Figure 2), enabling straightforward re-implementation.
> - Complete hyperparameter specifications for all FFN and Attention layers (Section 4.1), allowing for a faithful replication of our results.
> - Implementation details and pseudo-code for our custom sparse computation kernels (Appendix B), designed for direct integration into open-source frameworks like gemma.cpp.
>
>
> **Comparison with Alternatives (Table D.1) and FLOPS vs. Wall-Clock Time**
>
> Thank you for these points. Regarding comparisons with prior methods, our results in Figure 1(a) show that Spark achieves a greater FLOP reduction than methods like ReLU, ReLU^2, and Top-k, while maintaining comparable model quality.
>
> On the topic of evaluation metrics, we agree that direct wall-clock comparisons are valuable. However, fairly implementing and optimizing each baseline within our specific hardware and software framework (gemma.cpp) would be a significant engineering effort beyond the scope of this work. Therefore, to ensure a standardized and hardware-agnostic comparison against a wide range of published methods, we adopted FLOPs as our primary metric. This is an established practice in the activation sparsity literature (e.g., "ReLU strikes back"), allowing for more direct and fair comparisons with prior art.
>
>
> **GPU Evaluation on T4 vs. A100**
>
> We thank the reviewer for the opportunity to clarify our hardware selection. Our choice of the NVIDIA T4 was strategic, aimed at demonstrating the practical benefits of the Spark Transformer beyond high-end data center accelerators. While prior work has validated activation sparsity on premium hardware like A100s and TPUs, our goal was to prove its effectiveness on more widely accessible GPUs.
>
> Our strong results on the T4, together with our CPU performance, show a clear path toward making powerful LLMs more efficient on commodity and resource-constrained hardware. Therefore, our findings do not simply replicate prior work but instead complement it, by confirming that these efficiency gains are achievable across a broader and more common class of hardware.

---

> > ### Comment · Reviewer_4xrP · 2025-08-01
> >
> > Thanks for these answers. I would like to clarify my perspective on the paper. I recognize that Spark represents a strong technical contribution, presenting novel ideas and a variety of ablations, and the authors’ responses and the other reviews have reinforced this idea. I have no concerns regarding the motivation, technical soundness, or theoretical aspects of the work.
> >
> > My main concern is that, based solely on the manuscript, it is not entirely clear whether the paper provides sufficient scientific value. The work was clearly developed in an industry context, as all experiments are conducted with the proprietary Gemma codebase. In this setting, focusing on comparisons with the base Gemma model is reasonable, since industry projects typically target specific devices, datasets, and architectures. I also acknowledge that many existing approaches are poorly transferable between models and settings, and reproducing them can be impractical, particularly when baselines for Gemma 2 (e.g., with Deja Vu) do not exist. In isolation, the proposed method is interesting, well-motivated, and performant, and the paper itself is well written.
> >
> > At the same time, I believe that a scientific paper should clearly position the proposed method within the context of existing approaches in the literature. In this work, Spark is compared to other methods primarily through FLOPs and their theoretical properties (Table D.1), without reporting downstream speedups, despite such speedups being a core motivation for sparsity-based approaches. As a result, it is difficult to fully assess the practical impact of the method. Reporting FLOPs, as the authors mention, is common practice, but FLOPs alone are not always a reliable predictor of real performance gains, as downstream speedups depend heavily on hardware and implementation details. For example, Fig. 1 shows that while Spark reduces FLOPs to 40% of the base model, inference times only decrease to 50–60% for prefill and decode. Similarly, Fig. 5 illustrates that decoding speeds achieved with Spark vary considerably across hardware. Therefore, while I am fairly positive that Spark provides a good contribution as an efficient architecture, the absence of a fair downstream speed comparison with existing methods remains a significant limitation in fully evaluating the method’s impact.
> >
> > For example, it is not clear whether a reader can confidently conclude that Spark would outperform methods such as CATS when both are applied to a common 8B model like Llama, and I fail to get an intuition regarding the relative performance of Spark and other alternatives. As a result, the idea exists somewhat in a vacuum: Spark clearly provides acceleration relative to the base Gemma model, but I am not convinced that other existing approaches are not better choices- there is simply no data in the paper itself to conclude that.
> >
> > Overall, I have mixed feelings about the paper. On one hand, it is a strong technical contribution with novel ideas, solid motivation, and well-executed experiments within its industrial context. To my knowledge, some of the proposed changes are already integrated into 3n versions of Gemma, which further validates the relevance and importance of these contributions. On the other hand, I am not convinced that the paper demonstrates sufficient scientific rigor, primarily due to the lack of downstream speed comparisons with existing methods, which makes it difficult to fully assess its impact relative to the literature.
> >
> > I do not want to come across as an overly negative reviewer, especially since the paper is clearly impactful and well-received by the other reviewers. I don't feel confident enough to fully assess the trade-off between its strong practical impact and what I perceive as a lack of scientific rigor. Therefore, I will maintain my borderline reject score primarily to alert the AC and other readers to the evaluation concerns I have raised. However, I do not feel strongly about rejecting the paper and would be comfortable if it were accepted. Disregarding my concerns about the evaluation, the paper is well written, presents clear contributions, and would otherwise deserve a clear acceptance. I leave it to the AC to weigh these factors in the final decision.

---

> > > ### Author Response · Authors · 2025-08-01
> > >
> > > We are very grateful to the reviewer for this thoughtful and constructive follow-up. We deeply appreciate them taking the time to articulate their perspective, and respect the reviewer’s pursuit of scientific rigor. We agree that the trade-off between practical impact and academic evaluation standards is a crucial point of discussion, and would like to offer our perspectives.
> > >
> > > ## Paper contribution
> > >
> > > The primary scientific contribution of this paper is solving a critical prerequisite problem: **how to reactivate high levels of activation sparsity in modern, non-ReLU-based Transformers without the significant quality degradation that has plagued prior attempts.**
> > >
> > > This contribution is significant because the lack of natural sparsity in models like Gemma, Llama, and Mistral has become a major roadblock. It has prevented the community from applying the powerful efficiency techniques developed for older, naturally sparse models (like those discovered and targeted by MoEfication, Lazy Neuron, and Deja Vu). Our work unblocks this path. By demonstrating a method to achieve high sparsity (92% in FFN) with near-neutral quality, Spark provides a foundational architectural blueprint that enables a new generation of systems optimizations (e.g., custom kernels, specialized hardware) for the latest models. We believe this enabling role, in itself, constitutes a significant scientific value.
> > >
> > > ## Evaluation protocol
> > >
> > > It is highly uncommon in papers on activation sparsity to directly compare wall-time speedup against prior art. For instance, CATS [45], which the reviewer mentioned, primarily compares its speedups against its dense baseline, as does much of the related work. Even dedicated systems papers like PowerInfer focus on demonstrating speedups over dense baselines rather than providing an exhaustive wall-clock comparison against other sparse systems. As the reviewer rightly acknowledges, this shared practice exists precisely because of the immense difficulty in creating a fair comparison across different software and hardware stacks. Our methodology of comparing FLOPs more broadly while providing direct wall-time speedups against our primary baseline is consistent with community norms. This practice has not been holding the community back from introducing new ideas, developing valuable insights, and advancing the field forward.
> > >
> > > Finally, we thank the reviewer for their positive assessment of our work's technical novelty, motivation, and practical impact, including the acknowledgment that part of it is already integrated into the latest Gemma-3n. We hope this discussion helps frame the scientific contribution of our work more clearly. We appreciate the reviewer's candor and are confident in the value our paper brings to the NeurIPS community.

---

### Official Review · Reviewer_hAy9 · 2025-07-03

**Clarity:** 3
**Significance:** 3
**Originality:** 3
**Rating:** 5
**Confidence:** 4

**Summary:**

This paper introduces the Spark Transformer, a novel architecture designed to re-enable activation sparsity in both feedforward networks (FFNs) and attention mechanisms of Transformer models. The authors propose a statistical top-k operator—an efficient, differentiable approximation to top-k masking that avoids expensive sorting. They also design a unified predictor framework using reallocated parameters to estimate activation importance without additional overhead. The Spark Transformer achieves high sparsity (e.g., 8% FFN activation, ≤256 attended tokens), significantly reducing FLOPs while maintaining near-baseline quality. Extensive experiments on Gemma-2 show 1.4x decoding speedups on GPU, demonstrating the practicality and efficiency of their approach.

**Questions:**

No other questions.

**Ethical Concerns:**

["NO or VERY MINOR ethics concerns only"]

**Final Justification:**

The methodology and results are clearly presented, and the paper adds value to ongoing research in the field. I will therefore keep my score as Accept.

**Limitations:**

Yes

**Paper Formatting Concerns:**

yes

**Quality:**

3

**Strengths And Weaknesses:**

**Strengths**:
1. The Spark Transformer introduces a unified architecture for both FFN and attention sparsity using reallocated model parameters as low-cost predictors, avoiding extra modules or parameters and maintaining training simplicity.
2. The proposed statistical top-k operator provides an efficient, differentiable, and hardware-friendly alternative to traditional top-k methods, enabling sparse activation without significant training overhead.
3. The model achieves significant FLOPs and wall-time reductions (up to 1.79× on CPU, 1.4× on GPU) while maintaining competitive accuracy on standard downstream benchmarks compared to dense baselines like Gemma-2 2B.

**Weaknesses**:
1. The experiments are conducted solely on the Gemma-2 2B model, leaving it unclear how well the proposed method scales to larger models or generalizes to more complex scenarios such as instruction-tuned, or multimodal settings.
2. The paper introduces statistical top-k and shows that it outperforms standard top-k in empirical results. However, the paper lacks an in-depth analysis of why statistical top-k performs better. Is the improvement due to better differentiability? This is particularly important since the non-differentiability of sparse top-k has long been a bottleneck in training sparse MoE models, with many works proposing improved gating mechanisms. If statistical top-k indeed provides both performance gains and differentiability, it would be valuable to explore its potential applicability to MoE training, which the paper does not address.

---

> ### Author Rebuttal · Authors · 2025-07-31
>
> We thank Reviewer hAy9 for their positive assessment and for recognizing our work as "technically solid" with "high impact." We have found the comments very helpful in strengthening the paper and have revised the manuscript accordingly.
>
> # **Generalizability and Scaling (Evaluation on Gemma-2 2B only)**
>
> We agree with the reviewer that evaluating scalability to larger models and generalizability to instruction-tuning are valuable directions for future research. We have clarified this in the paper's conclusion.
>
> Our primary goal in this work was to provide a **rigorous and foundational study** of the Spark Transformer's effectiveness. To this end, we deliberately focused our significant computational resources on the Gemma-2 2B model for two key reasons:
>
> - **Controlled and In-Depth Evaluation**: Concentrating on a single, state-of-the-art open model allowed for a thorough and controlled study. This approach is a common and necessary practice in resource-intensive architecture research before undertaking even costlier scaling experiments.
>
> - **Strong, Representative Evidence**: Demonstrating significant FLOP reduction and wall-time speedup on a strong baseline like Gemma-2 provides **compelling evidence** for our approach's viability, establishing the groundwork for future scaling law analysis.
>
>
> # **Analysis of Statistical Top-k and Applicability to MoE**
>
> This is an excellent and insightful point. As detailed in the paper, the key advantages of our method are twofold:
>
> - **Advantages over Standard Top-k**:
>   - Efficiency: As shown in Figure 6, its linear-time complexity avoids costly sorting operations, yielding a >10x reduction in training overhead compared to optimized approximate top-k operators.
>   - Differentiability: Our use of soft-thresholding (Theorem 3.2) provides a smooth gradient signal, which we hypothesize leads to the superior convergence and final model quality we observed.
>
> - **Application to MoE Models**: This is a fantastic connection, and we thank the reviewer for suggesting it. Motivated by this comment, we investigated its applicability and found it to be a highly promising direction.
>   - A recent paper [a] uses a similar soft-thresholding technique for MoE routing but still relies on a costly sorting operator. Our method's ability to find the threshold in linear time could therefore address a key computational bottleneck, especially as MoEs trend toward a larger number of experts [17, 32].
>   - We believe this is an exciting avenue for future work and, to reflect the value of this insight, we will add a new discussion on this topic to the revised manuscript.
>
> [a] REMOE: ReMoE: Fully Differentiable Mixture-of-Experts with ReLU Routing, ICLR 2025

---

### Comment · Area_Chair_zErV · 2025-08-02
**The discussion phase has begun!**

Dear reviewers,

The rebuttal phase has now concluded. I encourage you to take a moment to carefully read the authors' responses and consider whether they address your concerns. Your insights during the discussion phase are important for reaching a fair and informed decision. If your opinion has changed based on the response or your reviews, please share your updated thoughts.

Thank you again for your time and contributions!

Best regards, Your Area Chair zErV.

---

### Note · Authors · 2025-08-14

We sincerely thank the AC and all reviewers for their insightful feedback, which has significantly strengthened our paper. We are encouraged that the reviews reached a consensus on two key points:

1.  **Novelty and Impact:** There is broad agreement (hAy9, PzjZ, 1Jfq) on the technical novelty of our statistical top-k operator and unified sparsity framework, and its practical impact via significant speedups on common hardware.

2.  **Real-World Validation:** The review process surfaced the ultimate validation of our work. As Reviewer 4xrP noted, our contributions are already being integrated into next-generation Gemma models, calling this a clear sign of their "relevance and importance." This direct adoption by a major open model family offers definitive, real-world validation that complements our empirical benchmarks.

The discussion also helped clarify our evaluation context. Our work’s core contribution is foundational: we re-enable activation sparsity in modern non-ReLU Transformers (e.g., Gemma, Llama) where it was previously infeasible without severe quality degradation. Our evaluation protocol, which shows wall-clock speedups against a strong dense baseline, is therefore consistent with community norms for foundational systems work (e.g., CATS, PowerInfer) that "unblocks" future optimizations.

Finally, inspired by the discussion, we have already produced new results that are ready for the camera-ready version, including new CPU speedup benchmarks and a deeper analysis of synergies with speculative decoding, quantization, and MoE routing.

We are confident our work offers a foundational blueprint and a strong empirical basis for the community. Thank you for your time and careful consideration.

---

### Decision · Program_Chairs · 2025-09-17

**Decision:**

Accept (poster)

**Comment:**

The paper proposes the Spark Transformer, a new architecture that reintroduces activation sparsity into both feed-forward networks (FFNs) and multi-head attention of Transformer models. The key idea is to selectively activate only the most important neurons or attention edges during inference, guided by lightweight predictors that repurpose existing parameters. Before the rebuttal, this paper receives one negative comment. However, after rebuttal and discussion, the major concerns of all the reviewers are addressed, especially about the novelty. Thus I also recommend to accept this paper.